# Assessment of miRNAs as transcriptional regulators in respiratory syncytial virus infection through computational analysis and molecular docking studies

Mubashir Hassan[1], Muhammad Shahzad Iqbal[2], Muhammad Yasir[3], Wanjoo Chun[3], Zainab Yaseen[4], Saba Shahzadi[1], Mark E. Peeples[5], Andrzej Kloczkowski[1,6,7]*

1 The Steve and Cindy Rasmussen Institute for Genomic Medicine at Nationwide Children's Hospital, Columbus, Ohio, United States of America, 2 Department of Biochemistry, University of Okara, Okara, Pakistan, 3 Department of Pharmacology, Kangwon National University School of Medicine, Chuncheon, Republic of Korea, 4 Department of Biotechnology, Faculty of Science and Technology (FOST), University of Central Punjab, Lahore, Pakistan, 5 Center for Vaccines and Immunity, Abigail Wexner Research Institute, Columbus, Ohio, United States of America, 6 Department of Pediatrics, The Ohio State University, Columbus, Ohio, United States of America, 7 Department of Biomedical Informatics, The Ohio State University, Columbus, Ohio, United States of America

* Kloczkowski.1@osu.edu

## Abstract

Globally, RSV is a major contributor to severe lower respiratory tract infections among children. Despite the significant medical concern posed by RSV, efforts to develop effective vaccines and antiviral drugs have largely fallen short, with the exception of immune prophylaxis available only for specific high-risk infants. We employed a suite of computational tools to investigate the role of microRNAs in the host's response to RSV infection. miRanda and RNAHybrid were instrumental in predicting microRNA-mRNA binding sites. For a deeper structural analysis, MC-Fold and MC-Sym were used to predict the 3D structures of both the miRNAs and their target mRNAs. The interactions between these molecules were then studied through RNA-RNA docking, with the resulting poses evaluated based on binding affinities and interaction profiles. This analysis focused on twelve selected miRNAs and their binding to specific sites on RSV mRNA. Finally, molecular dynamics (MD) simulations were conducted to evaluate the stability of the docked complexes. Taken together, these results suggest that two miRNAs, hsa_miR-2278 and hsa_miR-6732-3p, could potentially regulate the transcriptional activity during RSV infection and may warrant consideration as therapeutic agents.

## 1. Introduction

Respiratory syncytial virus (RSV) is a highly transmissible and prevalent virus from the Pneumoviridae family that causes respiratory tract and lung infections [1]. RSV is an

**Data availability statement:** Data Availability Statement: The data used or analyzed in the current study are available at different online resources including https://www.ncbi.nlm.nih.gov/; http://www.mirbase.org/; http://bibiserv.techfak.uni-bielefeld.de/rnahybrid; https://tools-4mirs.org/software/target_prediction/miranda/; https://major.iric.ca/MC-Fold/; https://major.iric.ca/MC-Sym/; and https://rnacomposer.cs.put.poznan.pl/, respectively. The supporting data has been mentioned in the supplementary file.

**Funding:** The author(s) received no specific funding for this work.

**Competing interests:** Authors declare no conflict of interest.

**Abbreviations:** RSV: Respiratory Syncytial Virus, miRNAs: MicroRNAs, N: Nucleoprotein, M: Matrix, F: Fusion, SH: Small hydrophobic, L: Large, G: Glycoprotein, UTRs: Untranslated regions.

enveloped virus containing a single, non-segmented, negative-sense RNA genome, approximately 15,191 nucleotides [2–4]. The genome of RSV consists of 10 genes, which encode 11 proteins [5]. Prior reports showed that RSV occurs mainly in infants (especially premature infants) and small kids, particularly before the age of two [6]. In adults age, RSV again becomes a problem, as it is for immune-compromised individuals, where it can be deadly. RSV infection typically presents with symptoms such as congestion, runny nose, dry cough, low-grade fever, sore throat, sneezing, and headacheg [7].

MicroRNAs (miRNAs) are non-coding RNAs control the transcription/translation profiling based on sequence complementarity [8]. The human miRNAs bind to viral mRNA complementary sequences during RSV replication and interfere in viral protein synthesis and replication processes [9]. The viral genes encode different proteins which are located on the linear genome in this order: nonstructural proteins 1 (NS1) and 2 (NS2), nucleoprotein (N), phosphoprotein (P), matrix protein (M), small hydrophobic protein (SH), attachment glycoprotein (G), fusion protein (F), matrix 2 proteins (M2-1 and M2-2), and polymerase (L). These proteins plays one or more essential roles in cell signal induction or inhibition, in the genome's transcription and replication, or in the assembly and function of the virion translocation to uninfected target cells [10].

Our present work is based on a methodology we recently applied to study miRNAs as possible regulators of mumps and influenza virus infection [11,12]. The present research uses computational approaches to predict human miRNA targets in the RSV genome, which might inhibit RSV replication and hinder its spread to other cells. The potential miRNAs have been accessed from different online resources and evaluated through miRanda and RNAhybrid algorithms. Additionally, molecular docking simulations was utilized to check the miRNA-mRNA binding in order to determine the nucleotides involves in viral replication. In future work, miRNAs can be further studied and used to develop antiviral therapeutics to cure viral infections. Moreover, miRNA delivery through nanoparticles is an example that supports this idea [13].

## 2. Results and discussion

### 2.1. RSV genome organization

RSV is a negative-sense, filamentous enclosed, single-stranded RNA virus that is spherical in shape, with a diameter of roughly 150 nm and a length of roughly 15191 nucleotides, depending on the strain. The 10 genes that make up the RSV genome encode eleven different proteins that work together to coordinate the highly regulated processes of viral replication, assembly, and immune evasion. The RSV proteome is an example of a complex viral strategy for survival and propagation, comprised of nonstructural proteins that modulate host immune responses (NS1, NS2), structural proteins that facilitate viral entry and egress (G, F, M), and regulatory proteins that ensure efficient RNA synthesis (N, P, M2-1, M2-2, L) (Fig 1).

### 2.2. miRNAs and target sites predictions in RSV genome

#### 2.2.1. Target prediction using miRanda. To uncover potential miRNA targets, the miRanda algorithm looks for genomic sites for miRNAs based on three principles:

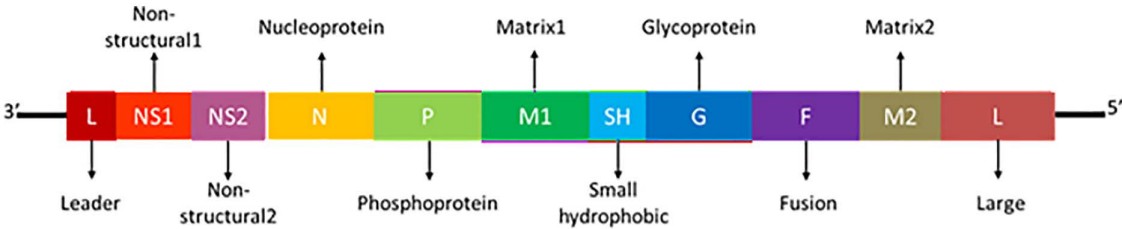

**Fig 1. RSV genome organization and proteins with length.**

binding energy, complementarity between sequences, and evolutionary conservation [14,15]. In our generated results, most of gene encoded proteins throughout the nucleotide sequences have been observed with energy range from (−20.0 to −25.0 kcal/mol), however, three proteins such as N, F and M showed greater energy values (−30.0 kcal/mol), respectively. The M protein showed maximum energy value in generated results. The predicted miRNAs binding to the genome sites that encode proteins F, G, L, M, M2, N, NS1, NS2, P and SH have been observed (Fig 2).

**2.2.2. RNAHybrid analysis.** RNAHybrid is a tool for determining the minimum hybridization free energy between a long and a short RNA that may be able to form a duplex with a particular human miRNA that is thermodynamically advantageous. Fig 3 shows that most targeted sites correspond to the MFE range from −25.0 to −32.0 kcal/mol, and nucleotides range from 0–15000 bp. Moreover, the top predicted gene-encoded proteins are F, G, L, M, M2, N, NS1, NS2, P and SH, respectively. In our predicted results, most of gene encoded proteins showed their existence on the target sites with energy value range from> −32.5 kcal/mol, whereas G protein exhibited highest free energy (Fig 3).

The comparative results showed that 48 and 312 miRNAs had been predicted with miRanda and RNAHybrid, respectively (Fig 4). All twelve top miRNAs listed in Table 1 were common for both miRanda and RNAHybrid predictions and may have excellent potential for binding to targeted sites in the RSV genome (Fig 4).

Moreover, in protein comparison results, we have identified multiple proteins F, G, L, M, N, SH, P, M2, NS1, and NS2 that may have a significant role in cell signaling and functionality of the RSV virus. Proteins G and F are the major glycoproteins on the surface of the virion and have essential roles in viral entry [10]. The twelve highest predicted miRNAs all have strong seed pairing interactions with specific portions of the RSV genome, which may be suggestive of a key regulatory role in viral RNA stability or translation. Their identification, therefore, could represent a valuable prelude to subsequent functional investigation and, ultimately, antiviral therapeutic development (Table 1).

## 2.3. Three-dimensional structures of miRNAs

The computational results generated from RNAHybrid and miRanda showed that twelve miRs were common predictions, which were further investigated by computing their docking energies. The predicted three-dimensional structure of all twelve human miRNAs (Table 1) has been shown in Fig 5.

**2.3.1. Small hydrophobic (SH) protein.** SH is a hydrophobic protein (65 AA) of RSV and is involved in viral infection [16] by inhibiting TNF-α production. Nevertheless, RSV without the SH gene remains alive, forms syncytium, and develops just as well as the wild-type virus, suggesting that the SH protein is not required for the virus to enter host cells [17]. It has been observed that, without the G protein, RSV expressing both F and SH proteins displays slightly smaller plaques, lower fusion activity, and slower viral entry than the virus that expresses F protein alone. Therefore, the SH protein harms virus fusion in cell culture [18].

**2.3.2. Nucleoprotein (N).** The N protein is made up of 391 amino acids, with an unstructured C-terminal tail and an N-terminal base structure [19]. The N-terminal domain is a major component responsible for inhibiting RNA, binding to P proteins, and nucleocapsid formation [20,21]. The fundamental function of the N protein for negative-strand RNA viruses

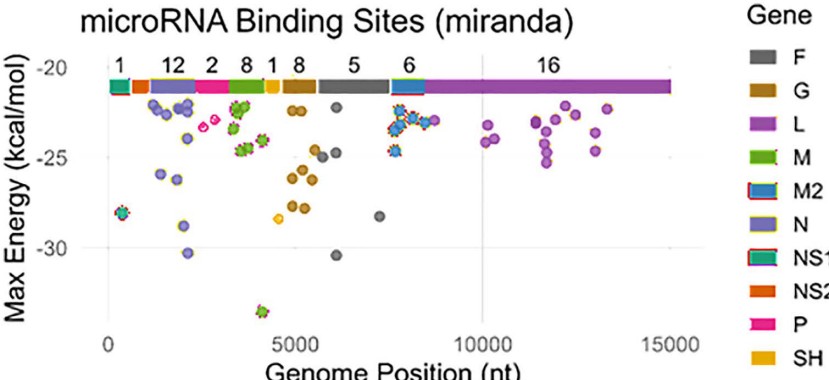

**Fig 2. Target sites/nucleotides positions of miRNAs in the genome of RSV predicted by miRanda algorithm.** The ggplot has been depicted between nucleotides on x-axis and energy values on y-axis. The list of different genes has been listed with different colors.

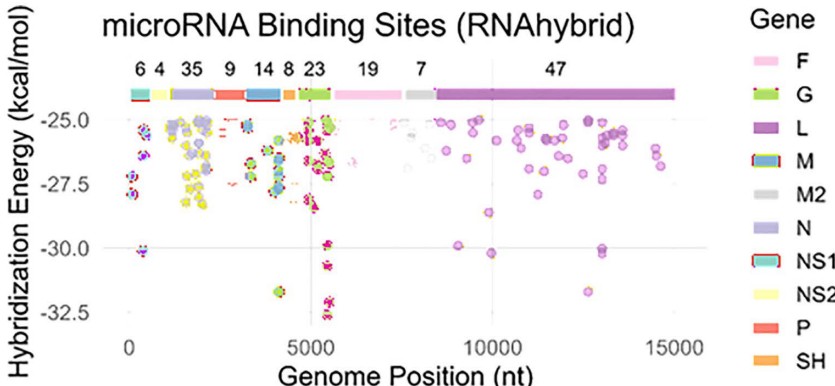

**Fig 3. Target sites/nucleotides positions of miRNAs in the genome of RSV predicted by RNAHybrid server.** The ggplot has been depicted between nucleotides on x-axis and energy values on y-axis. The list of different genes has been listed with different colors. The gray dots labeled as NA indicate nonapplicable cases.

is to construct the nucleocapsid of the virus by wrapping the viral genomic RNA during replication. The N protein typically forms a left-handed helical nucleocapsid that encloses the viral RNA genome and antigenome, shielding the RNA from nucleases [22–24].

**2.3.3. Fusion Protein (F).** The F protein mediates the fusion between viral and cellular membranes by converting the initial perfusion protein F1 to the extended intermediate in which four helices become one long heptad repeat (HR1) helix with the highly hydrophobic fusion peptide at its N-terminus that inserts into the target cell membrane [25]. The F1 subunit then folds like a closing jackknife to bring the two membranes together and trimerizes to become the core of the 6-helix bundle and the HR2 helices at the base of the F1 subunits pack against hydrophobic grooves of the HR1 trimers [26], bringing the lipid membrane of the virion and the cell together and initiating membrane fusion.

**2.3.4.. M2 protein.** The *M2* gene encodes two proteins M2-1 and M2-2 [27]. The mRNA encoding the M2 proteins of RSV contains two open reading frames (ORFs). The ORF1 encodes the 22-kDa structural protein, M2-1, and ORF2 encodes the 10-kDa protein, M2-2 [28]. The M2-1 protein contains a Cys3-His1 motif near the amino terminus, which is highly conserved among the M2-1 proteins of the members of the *Pneumoviridae* family [27]. The M2-1 protein acts

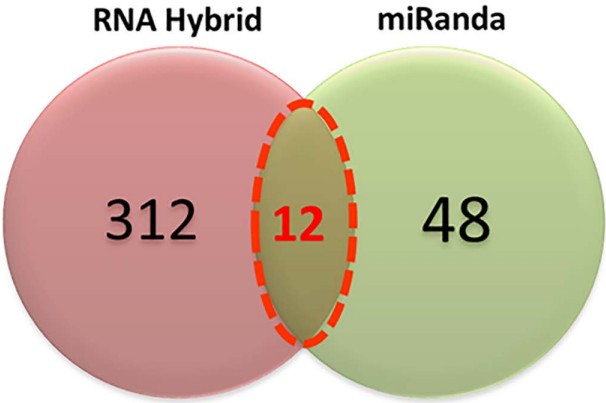

**Fig 4. A Venn diagram of predicted miRNAs by RNAHybrid and miRanda tools.** Twelve miRNAs were jointly predicted by both computational tools.

**Table 1. Precursor and Mature Sequences of miRNAs.**

| miRNAs | Precursor sequences | Mature sequences |
|---|---|---|
| hsa_miRNA_2278 | GUGCUGCAGGUGUUGGAGAGCAGUGUGUGUUGCCUGGGGGACUGUGUG-GACUGGUAUCACCCAGACAGCUUGCACUGACUCCAGACCCUGCCGUCAU | GAGAGCAGUGUGUGUUGCCUGG |
| hsa_miRNA_6081 | CCACCACGGUGCUGGCACCAGGGCCUCUGCCCCGUAGGACACCGAGGCU-UAUGAAUAGGAGCAGUGCCGGCCAAGGCGCCGGCACCAUCUUGGUGAU | AGGAGCAGUGCCGGCCAAGGCGCC |
| hsa_miRNA_4327 | GGCCUGGGUAGGCUUGCAUGGGGGACUGGGAAGAGACCAUGAACAG-GUUAGUCCAGGGAGUUCUCAUCAAGCCUUUACUCAGUAG | GGCUUGCAUGGGGGACUGG |
| hsa_miRNA_8089 | AAGGAGCACUCACUCCAAUUUCCCUGGACUGGGGGCAGGCUGCCAC-CUCCUGGGGACAGGGGAUUGGGGCAGGAUGUUCCAG | CCUGGGGACAGGGGAUUGGGGCAG |
| hsa_miRNA_6732_3p | AGGCCUAGGGGGUGGCAGGCUGGCCAUCAGUGUGGGCUAACCCUGUC-CUCUCCCUCCCAG | UAACCCUGUCCUCUCCCUCCCAG |
| hsa_miRNA_4761_3p | GGACAAGGUGUGCAUGCCUGACCCGUUGUCAGACCUGGAAAAAGGGC-CGGCUGUGGGCAGGGAGGGCAUGCGCACUUUGUCC | GAGGGCAUGCGCACUUUGUCC |
| hsa_miRNA_3135a | UCACUUUGGUGCCUAGGCUGAGACUGCAGUGGUGCAAUCUCAGUU-CACUGCAGCCUUGACCUCCUGGGCUCAGGUGA | UGCCUAGGCUGAGACUGCAGUG |
| hsa_miRNA_502_5p | UGCUCCCCCUCUCUAAUCCUUGCUAUCUGGGGUGCUAGUGCUGGCU-CAAUGCAAUGCACCUGGGCAAGGAUUCAGAGAGGGGGAGCU | AUCCUUGCUAUCUGGGGUGCUA |
| hsa_miRNA_6858_5p | GUGAGGAGGGGCUGGCAGGGACCCCUCCAAGUUGGGGACGGCAGC-CAGCCCCUGCUCACCCCUCGCC | GUGAGGAGGGGCUGGCAGGGAC |
| hsa_miRNA_200c_5p | CCCUCGUCUUACCCAGCAGUGUUUGGGGUGCGGUUGGGAGUCUCUAAUA-CUGCCGGGUAAUGAUGGAGG | CGUCUUACCCAGCAGUGUUUGG |
| hsa_miRNA_204_3p | GGCUACAGUCUUUCUUCAUGUGACUCGUGGACUUCCCUUUGUCAUC-CUAUGCCUGAGAAUAUAUGAAGGAGGCUGGGAAGGCAAAGGGACGUU-CAAUUGUCAUCACUGGC | GCUGGGAAGGCAAAGGGACGU |
| hsa_miRNA_4633-5p | UGGCAAGUCUCCGCAUAUGCCUGGCUAGCUCCUCCACAAAUGCGUGUG-GAGGAGCUAGCCAGGCAUAUGCAGAGCGUCA | AUAUGCCUGGCUAGCUCCUC |

as a transcriptional anti-terminator, enhancing the viral RNA polymerase's capacity to read across intergenic junctions and transcribe lengthy viral mRNAs. It permits the polymerase to access the genes located at the 3' end of the genome that are the furthest from the promoter [29]. Furthermore, it has been discovered that the M2-1 protein increases the polymerase's processivity and functions as a transcription elongation factor. The M2 gene is necessary for the recovery of infectious RSV from a cDNA clone [28,30].

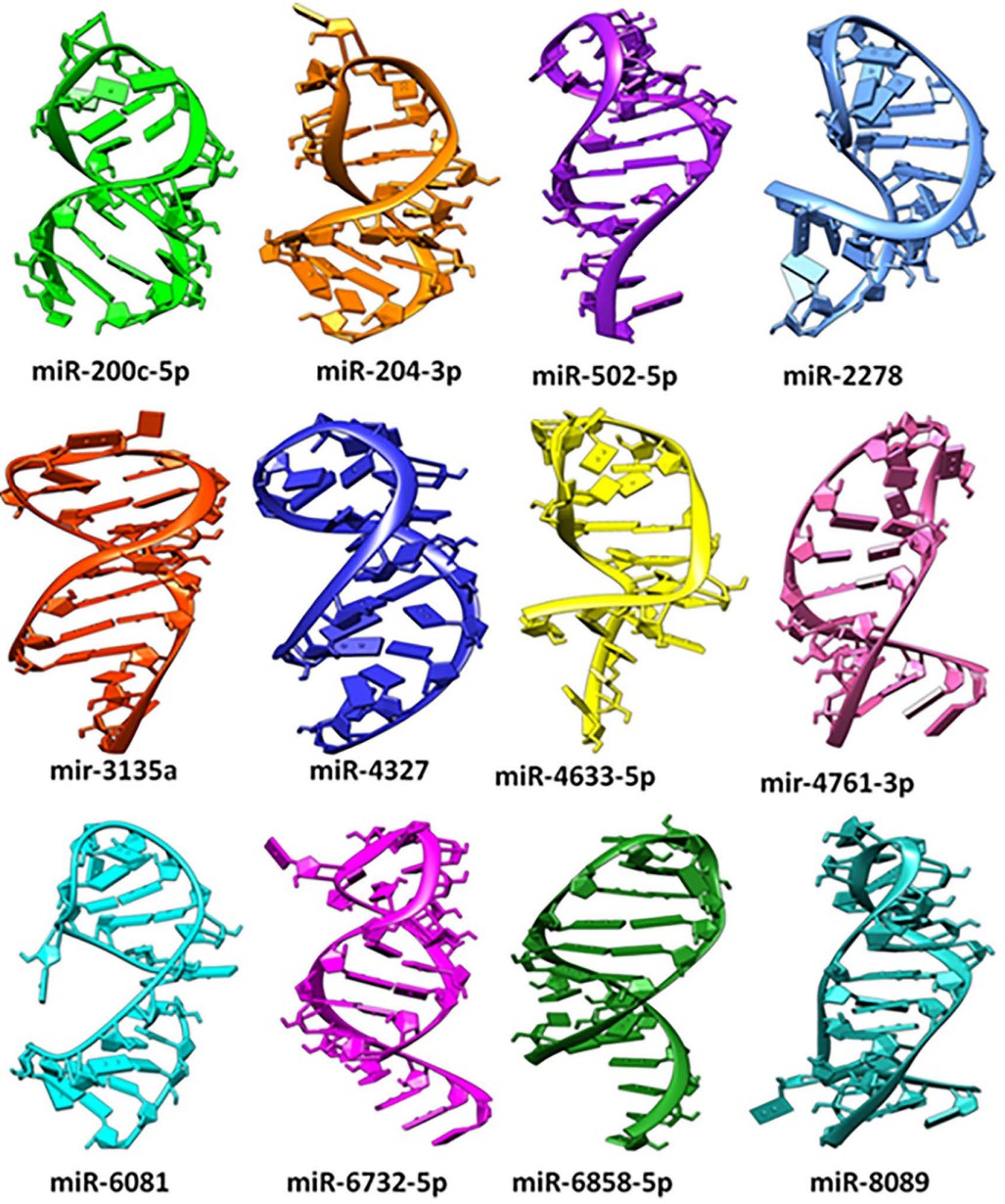

**Fig 5. Predicted structures of miRNAs.**

**2.3.5. Nonstructural protein 1 and 2 (NS½).** The NS1 protein (139 amino acids) is encoded by mRNA transcribed from the promoter-proximal RSV gene [31]. The helix α3 seen in the NS1 protein structure is thought to be important for NS1 functions and is largely conserved across RSV strains. Additionally, helix α3 is necessary for the protein's stability, IFN inhibition, and dendritic cell maturation suppression [32]. During the virus life cycle, NS1 is an antagonist of host type I and III interferon production, signaling, apoptosis, and dendritic cell maturation. It also controls protein stability and regulates transcription of host cell mRNAs, among other functions [33]. NS1 interacts with several host cell proteins to carry out a variety of tasks, although many of these proteins are linked to transcriptional regulation as a component

of the mediator complex that controls the cell cycle and other cellular processes [33]. NS2 is slightly smaller than NS1, containing 124 amino acids [31]. NS2 suppresses RIG-I-mediated IFN induction and early infection-related IRF3 activation. NS2 inhibits the RIG-I-MAVS connection and the subsequent activation of IRF3 by interacting with RIG-I via its N-terminal 229 AA. In order to support ongoing viral replication, NS2 suppresses apoptosis and IFN production and signaling [34–36].

**2.3.6. Large protein (L).** The RSV L protein is a single polypeptide of 2165 residues (250 kDa) and contains the catalytic core of RSV polymerase [37]. The L protein is a monomer with three functional domains: the RNA-dependent RNA polymerase, the capping enzyme (Cap), and the cap methyltransferase. Moreover, the L protein also possesses two other structural domains, the connector domain (CD) and the C-terminal domain (CTD) [38,39]. An RSV L polymerase inhibitor might provide a new tool for interrogation of the L protein function and could be helpful as an RSV therapeutic [40].

**2.3.7. Glycoprotein (G).** The G protein is a highly glycosylated membrane protein composed of 319 amino acids with an N-terminal intracellular cytoplasmic tail (1–37 AA), a transmembrane domain (38–66 AA), and an extracellular domain. The G protein is the most variable of the RSV proteins, so its sequence has been used to determine the subgroup and genotype of isolates [41]. The G protein mediates virion binding to target cells via heparan sulfate on cultured cell lines and CX3CR1 on ciliated cells in the respiratory tract [42]. G protein is also produced in a soluble form and secreted into the medium. Soluble G is generated by translation initiation at an internal in-frame AUG codon [43,44] and may disrupt the host immune response by binding to CX3CR1, a chemokine receptor on immune cells [45]. These features of G and its ability to block its function with antibodies directed at the CX3C region that likely binds to its in vivo receptor, CX3CR1, suggest that G has substantial potential for vaccine and antiviral drug design.

**2.3.8. Matrix protein (M).** The M protein is located external to the nucleocapsid layer, where it acts as a bridge between the lipid bilayer envelope and the nucleocapsid [46]. The M protein promotes the coordinated interaction between the structural elements of the virus to enable viral assembly by bridging the gap between the lipid bilayer envelope and the nucleocapsid [47]. The RSV M protein is believed to mediate the transportation of newly synthesized ribonucleoprotein complexes (RNPs) to assembly sites, which, in turn, drives assembly at the cell surface [47]. Moreover, M protein relate to its localization in the nucleus early in infection, although replication of the virus occurs in the cytoplasm [27].

## 2.4. mRNA-miRNA docking analysis

**2.4.1. The prediction of binding energy through HNA docking.** Molecular docking, a computational technique to check the binding interactions and energy/scoring values among biomolecules [48–50]. The predicted docking results show that miRNAs bind with the mRNA of RSV, forming various conformations with different scoring values. The predicted docking energy results showed that among twelve miRNAs, miRNA-6732-3p showed the highest scoring value (−383.68), whereas miRNA-3135a had the lowest scoring value (−141.38) (Fig 6). Moreover, the other generated docking complexes scored −150 to −300.

**2.4.2. mRNA-miRNA conformation and binding interactions analysis.** The anticipated docking complexes were carefully examined using mRNA-miRNA interactions, including hydrophobic and hydrogen bonding interactions. The knowledge of mRNA-miRNA interaction lies at the core of demystifying the regulatory networks of gene expression and holds therapeutic potential. Docking techniques facilitate this through providing a tool to predict and visualize the specificity and affinity of miRNA-cognate mRNA binding. The overall conformation of all docking complexes has been mentioned in Fig 7.

**2.4.2.1. miR-200c and miR204 interaction:** In miR-200c docking couple of hydrogen bonds have been observed with different nucleotides at different positions. The guanine (G) in miR200 at position 22 (G-22) interacts with cytosine (C) of mRNA at 106 (C-106) position having bond length 2.648Å. Whereas, another cytosine at position 13 (C-13) forms hydrogen bond with adenine at position 93 (A-93) having bond length 2.775 Å. Similarly, in miR-204-3p docking, the G at

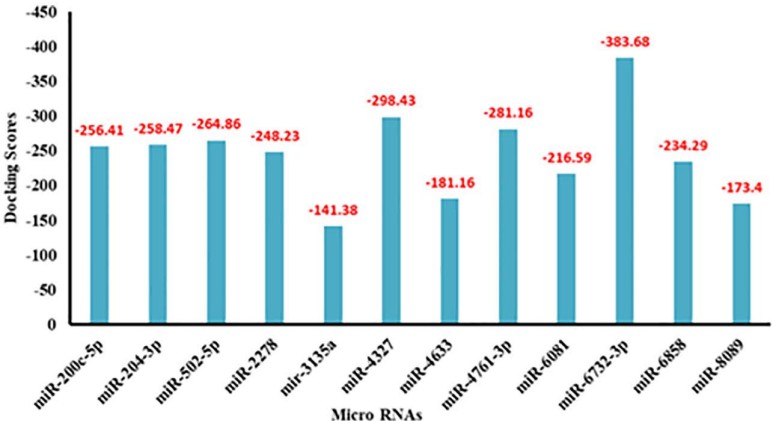

**Fig 6. Docking scores of predicted miRNAs using HNA docking server.**

positions 5 and 21 interacts with cytocine-99 (C-99) and adenine (A-100) with mRNA, having reasonable binding distances 3.137Å and 2.809Å, respectively. The interactions showed good strengthening in the complex formation (Fig 8).

**2.4.2.2. miR-502 and miR-2278 interaction:** The miR-502 docking investigation has revealed that three hydrogen bonds with distinct nucleotides at various locations. The adenine (A) in miR-502 at position 21 (A-21) interacts with adenine of mRNA at 34 (A-34) position having bond length 3.247Å. Whereas, uracil at position 17 (U-17) forms hydrogen bond with adenine at position 14 (A-14) having bond length 2.018 Å.

Furthermore, guanine at position 14 (G-14) forms another hydrogen bond with cytosine at position 147 (C-147) having bond length 3.099 Å. All three hydrogen bonds have appropriate bond distances in all three docking complexes and gave better strength and stability to docked complex.

Similarly, in miR-2278-3p docking, total six hydrogen bonds have been observed between miR and mRNA at different positions with appropriate bond lengths. The miR nucleotides such as guanine-5 (G-5), cytosine-6 (C-6), adenine-7 (A-7) and guanine-8 (G-8) showed good interaction against mRNA at different nucleotide positions. The guanine-5 (G-5) forms hydrogen bond with adenine 37 and 38 (A-37 & A-38) with bond lengths 2.971Å and 2.066Å, respectively. Furthermore, cytosine and adenine at positions 6 and 7 forms hydrogen bonds with mRNA nucleotide C-36 and uracil 48 (U-48) with bond distances 2.485 Å and 2.166 Å, respectively. Furthermore, G-8 form couple of hydrogen bonds with mRNA nucleotides at positions adenine 49 (A-49) and uracil 50 (U-50) with bond length 3.064 Å and 1.554 Å, respectively (Fig 9). The increased number of hydrogen bonds with appropriate bond distances form stability in the docking complex. Moreover, bond length, angle, and the chemical makeup of the contacting groups (nucleotide atoms) are some of the variables that affect the strength and stability of hydrogen bonds. However, to maximize the attractive forces between the donor and acceptor atoms, the ideal bond distance for hydrogen bonding is typically thought to be less than 3.5 Å [51].

**2.4.2.3. miR-3135 and miR4327 interaction:** The miR-3135-mRNA docking study has shown that five different nucleotides form hydrogen bonds with it at different places with appropriate bond length. The uracil at position 21 (U-21) in miR-3135 interacts with cytosine of mRNA at 91 (C-91) position having bond length 2.062Å. Similarly, uracil at position 16 (U-16) and guanine at position 7 in the miR-3135 form hydrogen bonds with guanine at position 107 (G-107) with bond length 2.177 Å and 2.912 Å. Moreover, the nucleotides with positions G-8 and U-106 showed interactions 2.747Å and 3.225Å, respectively. Most of H-bonds in this docking complex showed appropriate bond length and strengthen the interactive behavior. In miR4327-mRNA docking analysis, guanine and cytosine at positions 2 and 3 in miR form three H-bonds at two positions such as A94 and C95, respectively. The nucleotide G-2 form strong interaction with adenine at

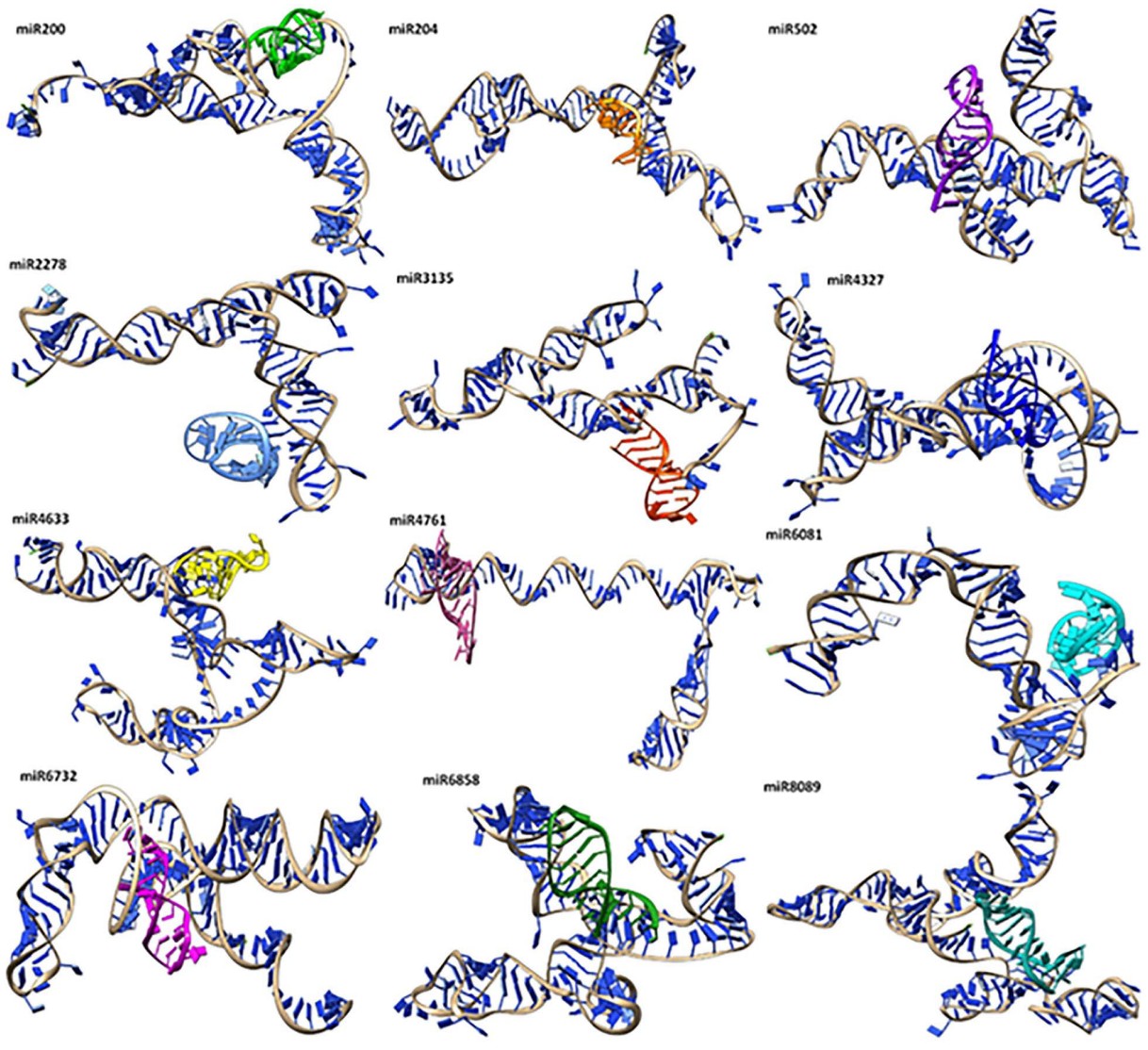

**Fig 7. Docking complexes of predicted miRNAs with mRNA.**

position 94 having bond length 1.263Å. Whereas, cytosine forms double hydrogen bonds with C95 with bond distances 3.202 and 2.731 Å, respectively (Fig 10).

**2.4.2.4. miR-4633 and miR4761-mRNA interaction:** In miR4633 and miR4761-mRNA docking analysis, there are different nucleotides at different positions forms hydrogen bonds with good binding lengths. The adenine at position 27 (A-27) in miR4633 form a hydrogen bond with cytosine at position 6 (C-6) in mRNA having bond distance 2.167Å. Whereas, guanine at positions 1 and 4, uracil at positions 15 and 19 in miR4761 actively participated in bind formation against the mRNA nucleotides. The guanine at positions 1 and 4 (G-1, 4) forms hydrogen bond with guanine 89 (G-89) and uracil at position 82 (U-82) having bond length 2.441 Å, and 2.500 Å, respectively. Moreover, uracil at positions 15 and 19 forms hydrogen bonds with adenine 83 and 85 with bond length 2.920 Å and 2.319 Å against the mRNA molecule (Fig 11).

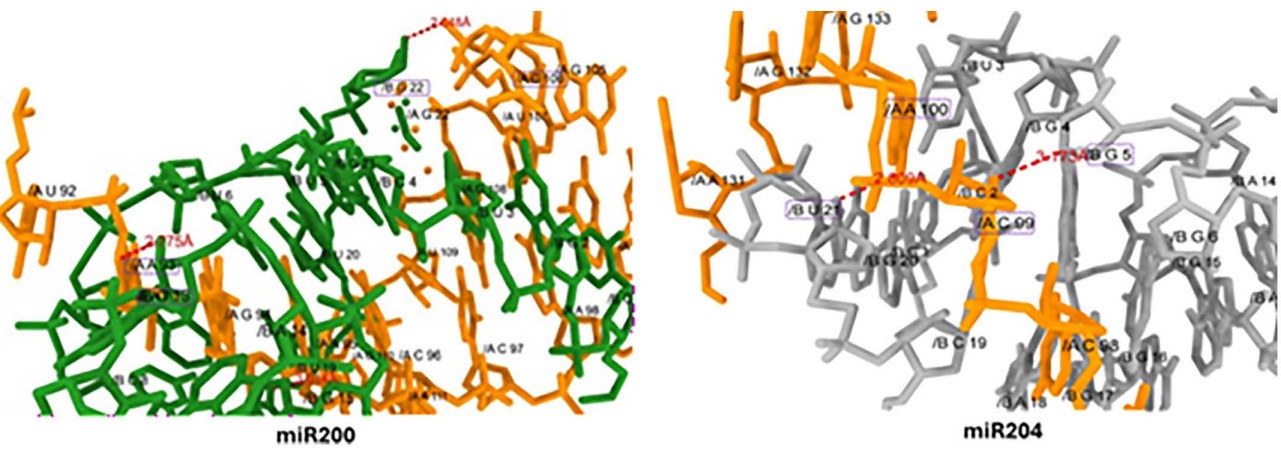

**Fig 8. Docking complexes miR-200c and miR204 interaction, the green and grey colors represent miRs whereas, dark orange color represents mRNA nucleotides.**

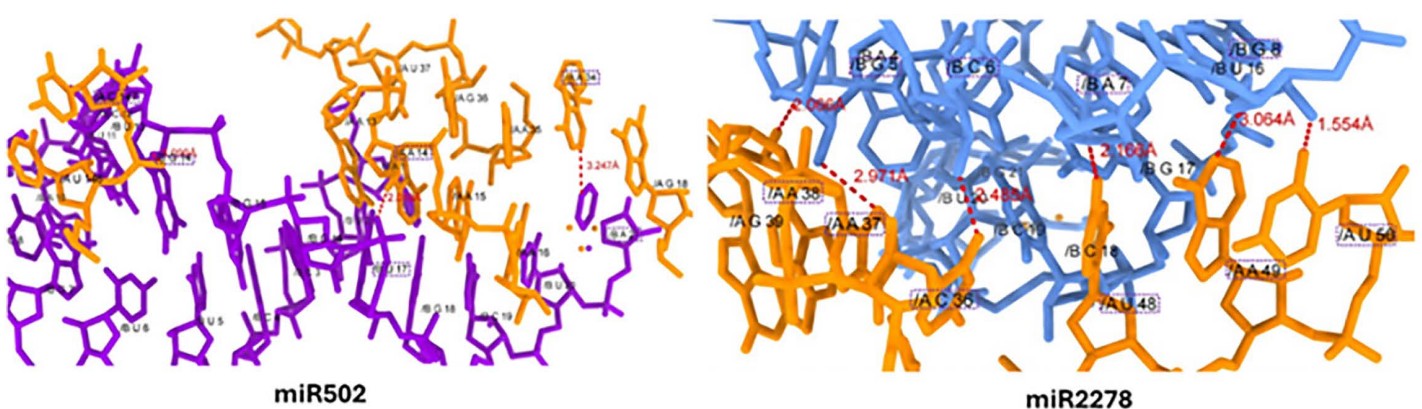

**Fig 9. Docking complexes miR-502c and miR2278 interaction, the purple and light blue colors represent miRs whereas, dark orange color represents mRNA nucleotides.**

**2.4.2.5. miR-6081 and miR6732-mRNA interaction:** In the docking analysis of miR6081 and miR6732-mRNA, distinct nucleotides at various locations establish hydrogen bonds with good binding lengths. The adenine at position 1 (A-1) in miR6081 form a hydrogen bond with guanine at position 76 (G-76) in mRNA having bond distance 2.727Å. Whereas, guanine at position 22 (G-22) forms a hydrogen bond with cytosine 53 (C-53) with bond length 2.325 Å. Both hydrogen bonds were comparable with standard H-bond value. Whereas, in miR6732-mRNA docking results, five hydrogen bonds have been observed at different nucleotides positions in both molecules. In miR6732, guanine at positions 1 and 3 (G-1 & 3) form hydrogen bonds with adenine at positions 17 and 18 (A-17 & 18) having bond length 3.188 Å and 2.821 Å, respectively. Moreover, cytosine 22 form hydrogen bonds with adenine 19 (A-19) having bond length 2.083 Å. Similarly, A-21 in miR6732 form two hydrogen bonds with adenine 20 (A-20) against mRNA molecule with bond distances 3.000 Å and 2.586 Å, respectively (Fig 12).

**2.4.2.6. miR-6858 and miR8089-mRNA interaction:** In the docking analysis of miR6858 and miR8089-mRNA, different nucleotides form hydrogen bonds with good binding lengths at different places. The cytosine at position 13 (C-13)

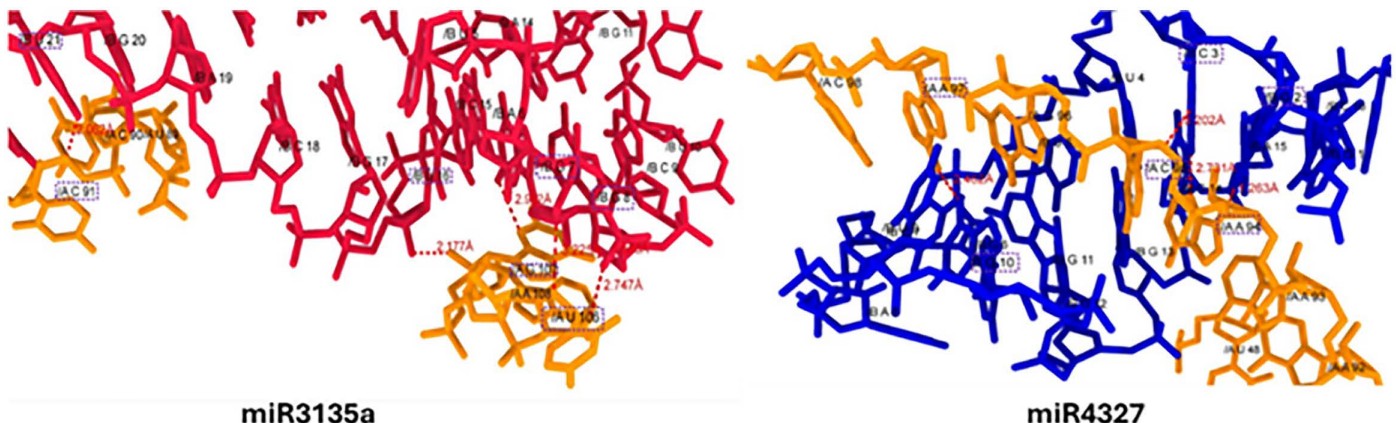

**Fig 10. Docking complexes miR-3135 and miR4327 interaction, the red and dark blue colors represent miRs whereas, dark orange color represents mRNA nucleotides.** The H-bonds represented by red dotted lines and the nucleotide involved in interactions are highlighted by dotted purple rectangles.

in miR6081 form a hydrogen bond with adenine at position 93 (A-93) in mRNA having bond distance 2.775Å. Whereas, uracil at position 18 (U-18) forms a hydrogen bond with adenine 95 (A-95) with bond length 3.014 Å. Moreover, guanine at position 22 (G-22) forms a hydrogen bond with cytosine 106 (C-106) with bond length 2.648 Å. Whereas, in miR8089-mRNA docking results, single hydrogen bond has been observed at different nucleotides positions in between both molecules. In miR8089, guanine at positions 4 (G-4) form hydrogen bond with uracil at positions 131 (U-131) having bond length 2.552Å. (Fig 13).

Therefore, these predicted miRNAs may also help reduce the pathogenicity virulence of the viral agents upon inhibiting F mRNA translation. Prior data also showed that miRNAs regulate the immune response against viral infections [52]. The prior research data explored that hsa_miRNA-6732-3p targets several genes related to cell signaling and death that could impact RSV persistence and replication in the host. Furthermore, differences in hsa_miRNA-6732-3p expression could shed light on the processes underlying immunological dysregulation brought on by RSV [53]. Numerous miRNAs exhibit dysregulation in their expression during infection, according to studies on the expression profiles of miRNAs in neonates infected with RSV. For example, it has been observed that nasal mucosal samples of patients with mild and severe RSV sickness exhibit significantly varied levels of hsa_miRNA-2278, highlighting its possible role in differential disease severity [54]. Another report showed that hsa_miRNA-2278 might be involved in regulating cellular reactions and inflammatory pathways, both of which are essential for efficient virus removal. The entire course of RSV disease can be influenced by the interaction between hsa_miRNA-2278 and viral determinants, which can either promote viral replication or lead to immune-mediated tissue damage [55].

## 2.5. Molecular dynamic simulation analysis

**2.5.1. RMSD.** In our 20-ns all-atom GROMACS simulations, each miRNA-target complex began with an initial backbone RMSD of approximately 0.5 nm and exhibited a clear upward trajectory across the corresponding bar-graph plots (Fig 14) before reaching their respective end-point values; notably, hsa_miRNA-3135a, hsa_miRNA-6732-3p, and hsa_miRNA-4327 showed the smallest net increases, rising steadily from 0.5 nm to final RMSDs below 1.0 nm, which underscores their exceptional rigidity and minimal conformational rearrangement over the full simulation. By contrast, hsa_miRNA-2278 and hsa_miRNA-200 demonstrated moderate upward trends from the 0.5 nm baseline to end-point RMSDs of ~1.4 nm by 20 ns, indicating a balance of induced-fit dynamics and eventual stabilization. A slightly greater degree of fluctuation was observed for hsa_miRNA-6081, −6858, −8089, and −4761, each traversing from 0.5 nm up to

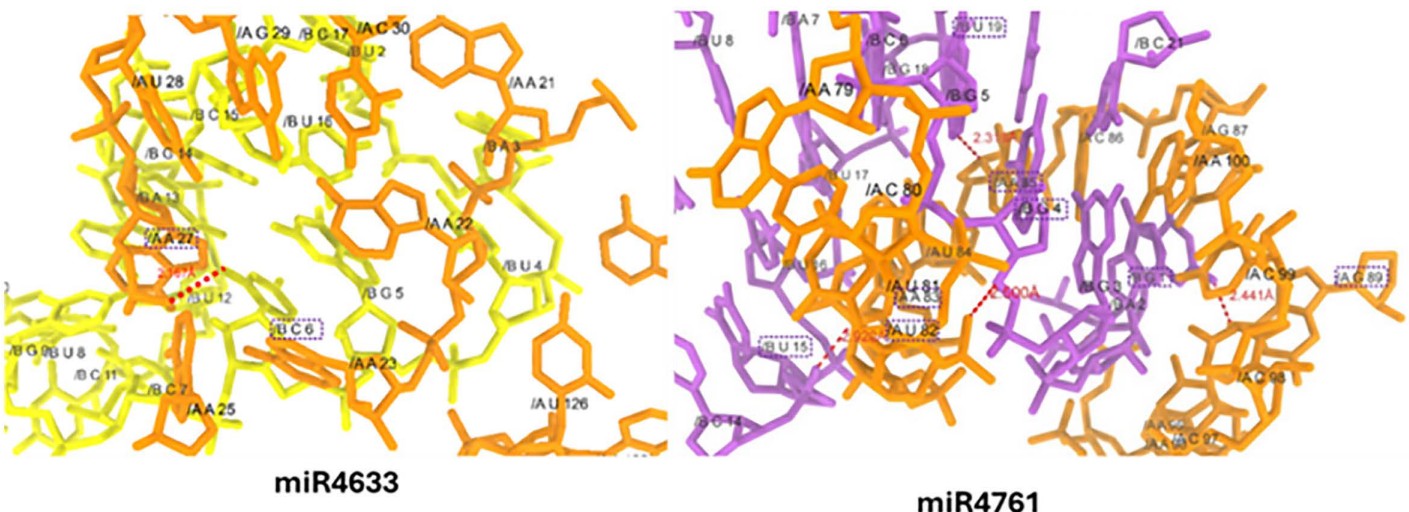

**Fig 11. Docking complexes miR-4633 and miR-4761.**

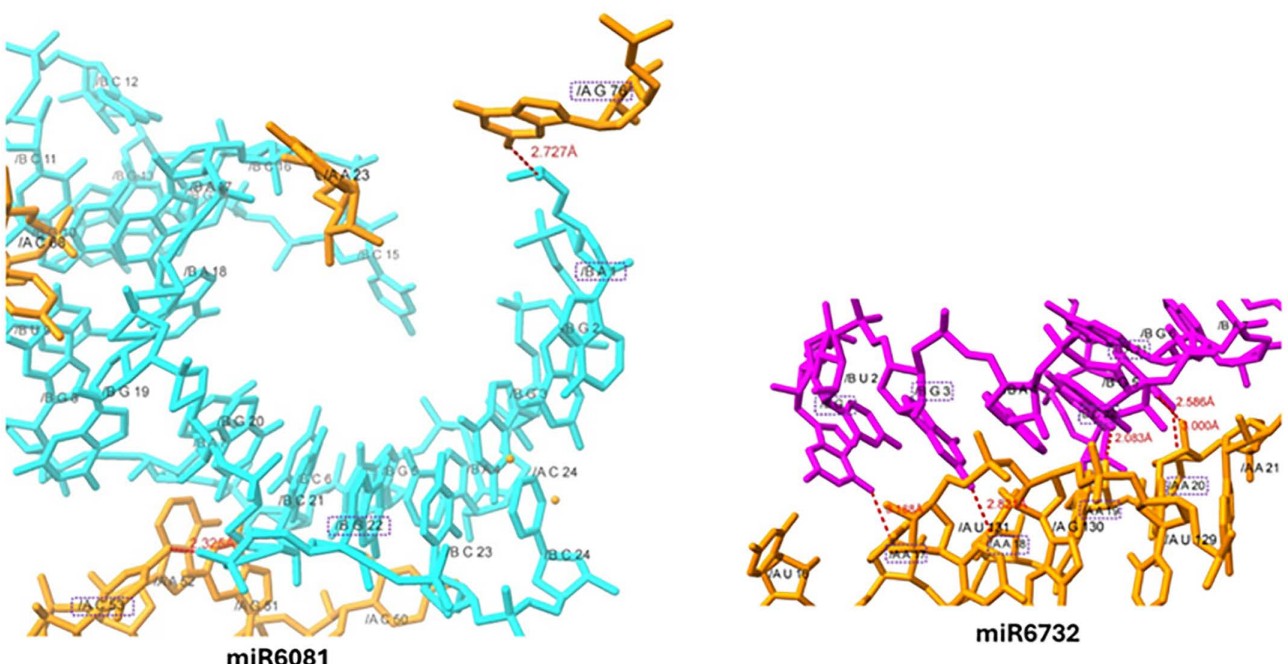

**Fig 12. Docking complexes miR-6081 and miR6732 interaction, the cyan and purple colors represent miRs whereas, dark orange color represents mRNA nucleotides.** The H-bonds represented by red dotted lines and the nucleotide involved in interactions are highlighted by dotted purple rectangles.

between 1.6 and 1.8 nm, reflecting enhanced backbone mobility yet a discernible plateau later in the trajectory. Finally, hsa_miRNA-204, −502, and −4633 exhibited the steepest upward slopes, climbing from 0.5 nm to RMSDs in excess of 2.0 nm without reaching a stable plateau by the end of the 20 ns window, signifying highly flexible, transient interactions.

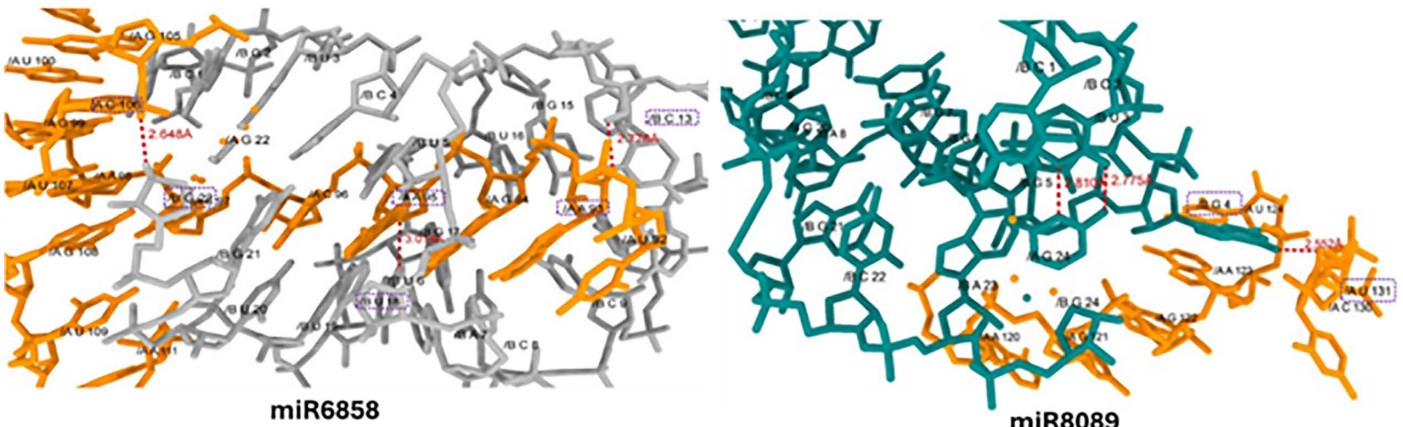

**Fig 13. Docking complexes miR-6858 and miR8089 interaction, the gray and dark cyan colors represent miRs whereas, dark orange color represents mRNA nucleotides.** The H-bonds represented by red dotted lines and the nucleotide involved in interactions are highlighted by dotted purple rectangles.

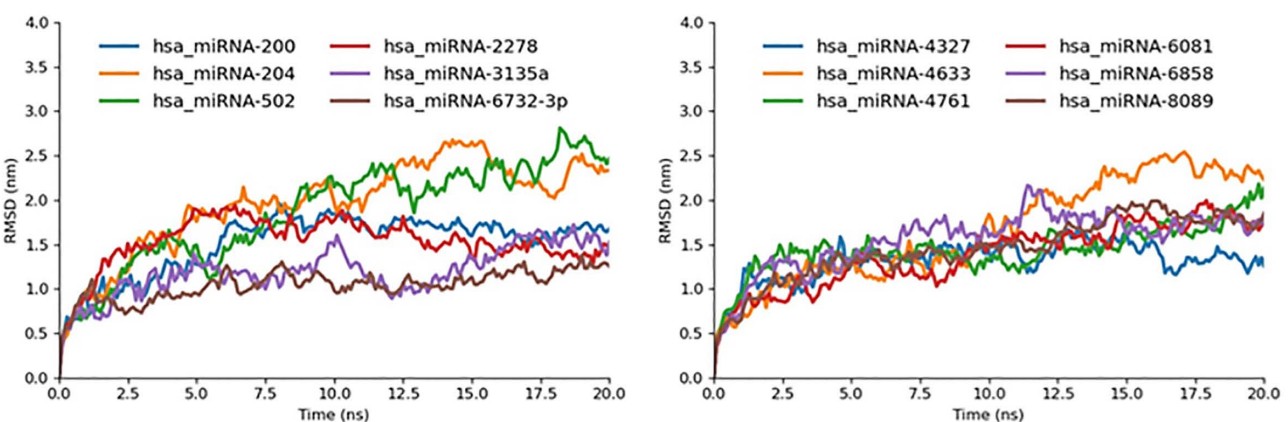

**Fig 14. RMSD graph miR-mRNA docked complexes represented in different colors.**

Overall, this comprehensive RMSD analysis permits us to categorize the complexes into four stability tiers-highly stable, moderately stable, intermediately flexible, and unstable thus prioritizing hsa_miRNA-3135a, −6732-3p, and −4327 for downstream experimental validation.

**2.5.2. RMSF.** To complement our global stability assessment by RMSD, we performed per-residue flexibility analysis via the Root-Mean-Square Fluctuation (RMSF) over the same 20 ns GROMACS trajectories (Fig 15). In general, most miRNA-receptor complexes exhibited similar backbone fluctuation profiles, but five complexes, hsa_miRNA-8089, −204, −502, −6858, and −4761, stood out with pronounced peaks exceeding 1.3 nm at loop and turn regions, indicating localized flexibility hotspots. Conversely, has-miRNA-2278 displayed the lowest overall RMSF, with maximum fluctuations of only ~0.9 nm across all residues, marking it as the most rigid complex at the local level.

The remaining complexes (has-miRNA-200, -3135a, −6732-3p, −4327, −4633, and −6081) showed moderate RMSF values (0.6–1.0 nm), consistent with balanced flexibility that can accommodate induced-fit interactions while maintaining a stable binding core. When these RMSF findings are compared with our earlier RMSD results, a clear concordance emerges: complexes with the lowest end-point RMSD values (3135a, 2278 and −6732-3p) also exhibit minimal

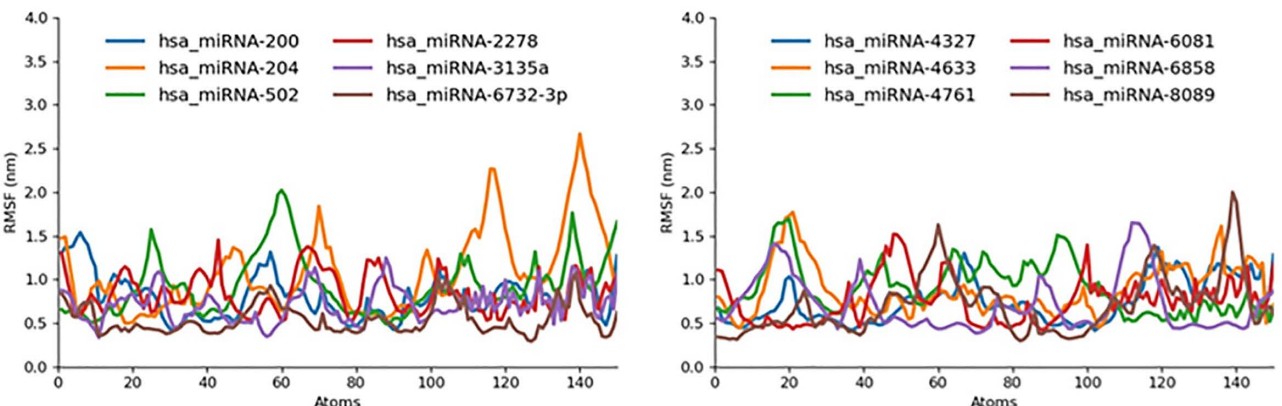

**Fig 15. RMSF graph miR-mRNA docked complexes represented in different colors.**

per-residue fluctuations, underscoring both global and local stability; in contrast, those with the highest RMSD deviations (8089 and 6081) likewise show elevated RMSF, reflecting overall instability coupled with flexible binding regions. Notably, hsa_miRNA-2278 presents an interesting case, with a moderate global RMSD (~1.4 nm) yet exceptionally low RMSF, suggesting that after an initial conformational adjustment, it achieves a particularly rigid interaction interface, making it a promising candidate for further experimental validation.

**2.5.3. SASA.** To quantify the extent of solvent exposure at the miRNA target interfaces, we computed the solvent-accessible surface area (SASA) over the 20 ns MD trajectories and found a clear stratification of the complexes into low, moderate, and high SASA groups. hsa_miRNA-3135a, hsa_miRNA-2278, hsa_miRNA-4761, and hsa_miRNA-6081 each exhibited the smallest mean SASA values, approximately 200 nm², indicating that these complexes bury more surface area upon binding and thus maintain tighter, more sequestered interfaces. At the other extreme, hsa_miRNA-8089 reached a peak SASA of ~340 nm², signifying a highly solvent-exposed interface that likely affords greater conformational flexibility but may compromise binding affinity. The remaining complexes hsa_miRNA-200, hsa_miRNA-204, hsa_miRNA-502, hsa_miRNA-6732-3p, hsa_miRNA-4327, hsa_miRNA-4633, and has_miRNA-6858 fell into an intermediate SASA range (280–300 nm²), reflecting a balance between interface burial and solvent accessibility. These SASA trends correlate well with our RMSD and RMSF analyses: the low-SASA group aligns with the most rigid, low-fluctuation complexes, while the high-SASA outlier (8089) corresponds to the highest RMSD/RMSF values, collectively underscoring how solvent exposure complements dynamic stability metrics in predicting the strength and resilience of each miRNA-target interaction (Fig 16).

**2.5.4. $R_g$.** To evaluate the overall compactness and folding stability of each miRNA target complex, we calculated the radius of gyration ($R_g$) over the 20 ns MD trajectories and observed distinct grouping: hsa_miRNA-3135a and hsa_miRNA-6081 exhibited the lowest average $R_g$ values (3.2–3.4 nm), indicating the most tightly packed, compact conformations; in stark contrast, hsa_miRNA-4761 reached the highest $R_g$ of up to 5.4 nm, reflecting substantial expansion and lower structural cohesion. The remaining eight complexes hsa_miRNA-200, −204, −502, −2278, −6732-3p, −4327, −4633, −6858, and −8089 clustered in a moderate $R_g$ range of 3.8–4.5 nm, signifying intermediate compactness. These $R_g$ trends align closely with our RMSD, RMSF, and SASA analyses: the low-$R_g$ complexes correspond to those with minimal fluctuations and solvent exposure, while the high-$R_g$ outlier mirrors the greatest instability and solvent accessibility, thereby reinforcing the identification of 3135a and 6081 as the most structurally robust candidates and 4761 as the least (Fig 17).

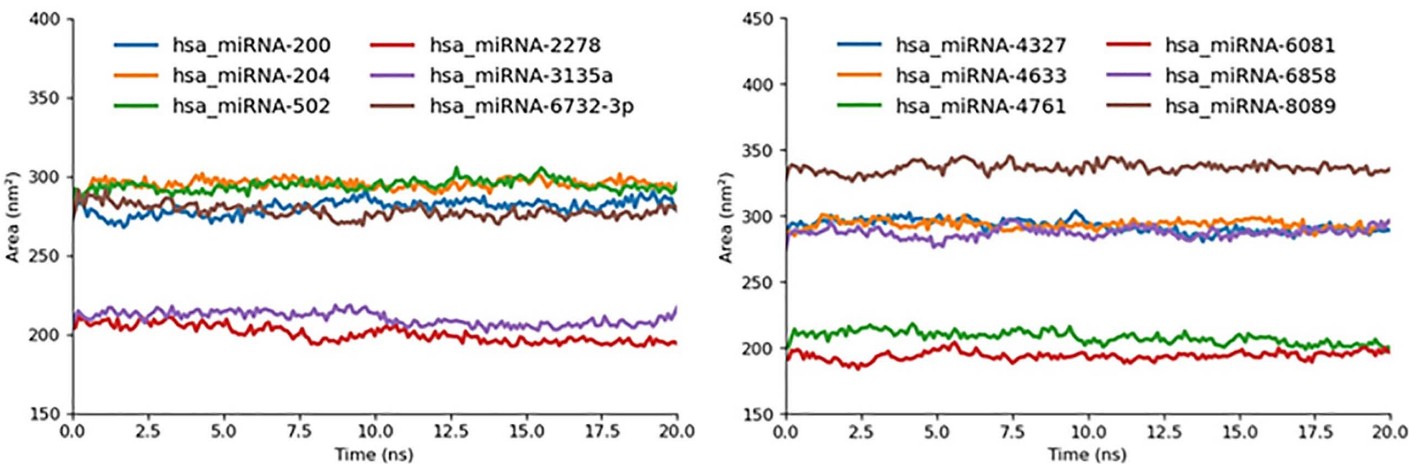

**Fig 16. SASA graph miR-mRNA docked complexes represented in different colors.**

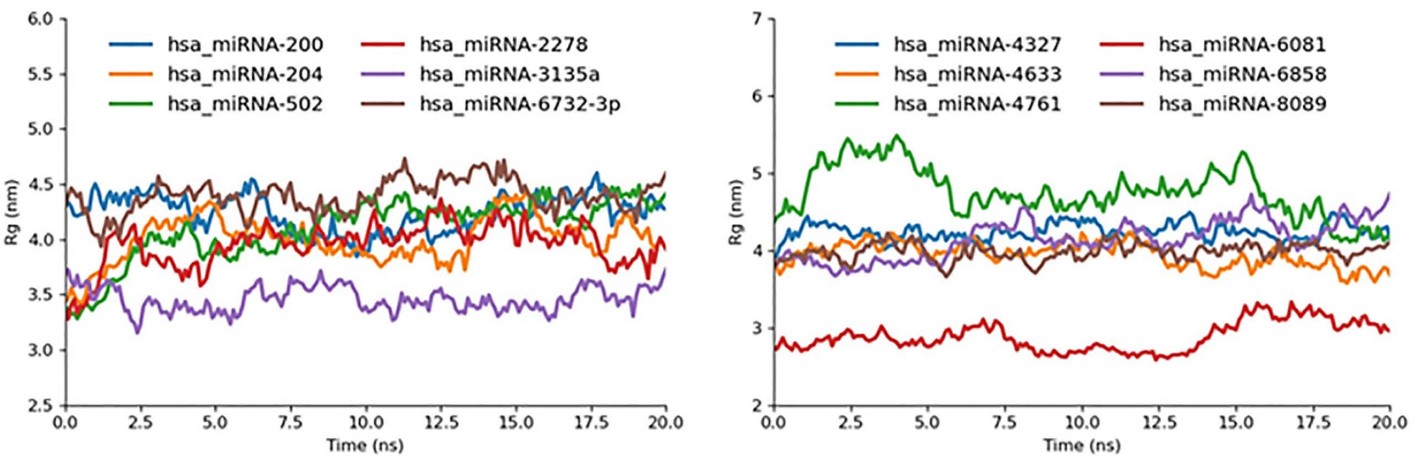

**Fig 17. Rg graph miR-mRNA docked complexes represented in different colors.**

## 2.6. miRNAs as transcriptional profiler in RSV infection

RSV is a non-segmented negative sense RNA virus belonging to the *Paramyxoviridae* family and is responsible for significant numbers of severe human infections each year. It has been observed that miRNAs regulate the responses related to both acquired and innate immunity of the humans by controlling the transcription profiling [56]. Respiratory droplets from an infected individual spread the disease if they encounter the mucosa of another individual's eyes, mouth, or nose. Once an individual is exposed, the viral and host cell membranes are fused. The glycoproteins F and G on the viral surface regulate viral attachment and the early phases of infection. The endoplasmic reticulum and exocytic route allow the surface glycoproteins to be produced, post-translationally altered, and delivered. Both the polymerase and the viral nucleocapsid enter the cytoplasm of the host cell. The viral genome is converted to mRNA by RNA-dependent RNA polymerase, and the host then translates it. To create negative-sense RNA, the RNA polymerase creates a complementary template strand

called the positive-sense antigenome. After being bundled into nucleocapsids, the resulting RNA is transported to the plasma membrane for budding and assembly (Fig 18). miRNAs represent a class of regulatory RNAs in host-pathogen interactions [57,58]. The prior data reported that novel immune-associated miRNA expression profiles in the nasal epithelium of RSV-positive infants which showed the down and upregulation of miRNAs. Furthermore, two miRNAs like miRNA-125a and miRNA-429 are only down-regulated in mild and not severe disease, and the lack of down-regulation in severe disease may explain the observed differences in disease manifestations following RSV infection [54,59,60]. In our predicted results, hsa_miRNA-2278 and hsa_miRNA-6732-3p have good binding potential with mRNAs of SH, P, L and N proteins which are responsible for RSV replication process. As a result, miRNAs can either destabilize mRNA or prevent its translation. Therefore, the altered functionality of these proteins may help in to control the replication process of RSV in the host cell and may help in the boosting the immune system. It thus appears that predicted miRNA (hsa_miRNA-2278 and hsa_miRNA-6732-3p) are playing a key role as a therapeutic target in the development and progression of RSV infection.

## 3. Computational methodology

### 3.1. RSV genome retrieval and annotation

The complete genome sequence of the Respiratory Syncytial Virus (RSV) with the accession number NC_001803.1 was downloaded from NCBI (https://www.ncbi.nlm.nih.gov/). CLC Genome Workbench (v 9.5.2) has been used to view and analyze sequences to obtain more information.

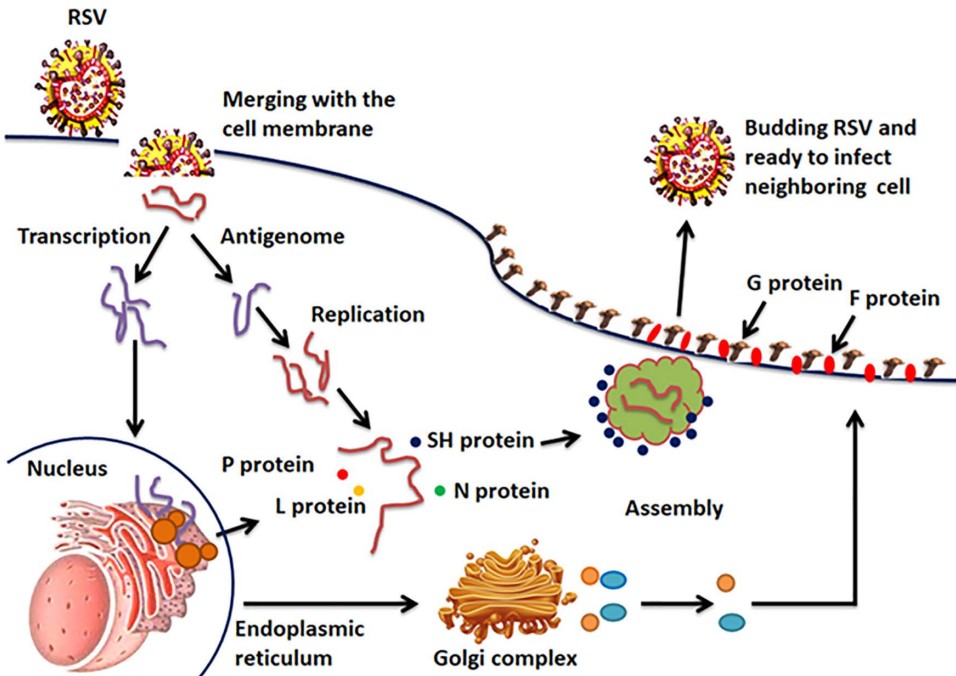

**Fig 18. The transmission and replication pathway of RSV. An infected person's respiratory droplets come into touch with another person's mouth, nose, or eye mucosa.** The membranes of the virus and host cells fuse together after an individual is exposed. Glycoproteins F and G on the viral surface regulate viral attachment and the early phases of infection. The polymerase and viral nucleocapsid enter the cytoplasm of the host cell. The viral genome is converted into mRNA by RNA-dependent RNA polymerase, and host proteins then translate this mRNA. In order to create negative-sense RNA, the RNA polymerase creates a complementary template strand called the positive-sense antigenome. Nucleocapsids with the resulting RNA are transported to the plasma membrane for assembly and budding.

## 3.2. Retrieval of mature miRNAs of humans

The 360 human miRNAs that have been identified were retrieved from miRbase (http://www.mirbase.org/), an online database having a collection of miRNAs from different species [61,62]. Each entry in the miRBase database displays information about the location and sequence of precursor and mature miRNA sequences in supporting information (S1 Data).

## 3.3. miRNA target prediction in the RSV genome

Two online tools miRanda [14], and RNAHybrid [63] were employed to predict miRNA targets of human-derived miRNAs against the RSV genome to locate miRNA targeting regions. The sequences of human miRNAs and the genome of RSV, each in FASTA format, were processed by these algorithms to examine a variety of parameters. miRanda is an algorithm used for both animals and plants [14]. In general, miRanda concentrates on fundamental factors including gap penalties, a weighted total of match and mismatch scores for base pairs, the free energy of the RNA-RNA duplex, cross-species conservation of the target site, and sequence complementarity. Additionally, it raises the specificity by enabling the prediction of several miRNA target sites, even those with poor binding in the seed region within the target site's 3′UTR [15,64]. The algorithm was run after defining the settings (gap open penalty = −9.0, gap extension penalty = −4.0, score threshold = 140, energy threshold = −18 kcal/mol, and scaling parameter = 4.0). The gap open penalty is the penalty for opening a gap in the sequence alignment. The increase in this value makes the gaps less frequent. The gap extension penalty is the penalty for extending a gap by one residue, and increasing this value will make the gaps shorter. The threshold score is a combination of sequence score and free energy that achieves the best performance. The algorithm scores alignments based on sequence complementarity rather than sequence identity. The scaling parameter in miRanda is used to match or mismatch scores in the 5' end of a microRNA [65].

RNAHybrid is another tool for finding the minimum free energy for long and short miRNA [63]. In domain mode, hybridization is carried out by combining the short sequence with the most suitable portion of the long sequence. miRNAs are short-acting RNAs that bind to certain mRNAs to control gene targeting [66]. Typically, RNAHybrid detects excellent hybridization sites of small RNA to large RNA. RNAHybrid default parameters (minimum free energy at −30.1) were used to predict the RSV's miRNA targeting sites.

## 3.4. Rstudio and Ggplot2

The R-language integrated development environment (IDE), RStudio (https://rstudio.com/), is typically used for mathematical computations and visualizations. The ggplot2 package (https://cran.r-project.org/web/packages/ggplot2/index.html) was utilized to illustrate the miRNA prediction graphical representations.

## 3.5. miRNA and protein retrieval

MC-Fold (https://major.iric.ca/MC-Fold/) and MC-Sym (https://major.iric.ca/MC-Sym/) are online platforms utilized for miRNAs model prediction. Whereas, mRNA model prediction was done by RNAComposer using default parameters [67]. UCSF Chimera 1.0 tool was used to visualize the 3D models of miRNA and mRNA and to depict their graphical images [68]. The predicted secondary structure of mRNA has been depicted in supporting information figure (S1 File).

## 3.6. Molecular docking of mature miRNA and mRNA

Molecular docking is an significant approach to check the interaction behavior among ligands and proteins [49,50]. The HNADOCK uses a hierarchical docking algorithm of an FFT-based global search strategy and an intrinsic scoring function to predict binding conformations and interactions among biomolecules [69]. HNADOCK server was employed to check its binding pattern with predicted genomic motifs and observe their conformational behavior. The miRNA-mRNA docking is based on a hybrid algorithm of template-based modeling and an ab initio-free docking method. In the general procedure,

3D structures of both mRNAs of RSV are predicted and run docking with maximum search space. The best docking complexes against each miRNAs were selected and downloaded based on scoring values using scoring function DITScoreRR [69] and analyzed in Discovery Studio (2.1) [70] and UCSF Chimera 1.10.1, respectively.

### 3.7. Molecular dynamics simulations

The standard parameters and the protocol of simulations were selected for the 20 ns MD simulation experiment of docked RNA complexes. The GROMACS program (version 2019.3) was used to examine the structural behavior of protein and ligand complexes using prior protocol [71]. The CHARMM-GUI server's solution builder protocol was used to generate the CHARMM36 force field, and the same interface was used to construct input files for MD simulations in GROMACS [72]. The TIP3P 3-point water model was utilized to solvate the system, a cubic box with periodic boundary conditions. Counterions were added until the system was neutralized. The Verlet algorithm, with a cut-off radius of 10 Å for electrostatic and van der Waals interactions, was employed while the LINCS algorithm was used to constrain the bond lengths during the simulations. Furthermore, the Particle Mesh Ewald (PME) method was used to calculate electrostatic interactions. To execute MD simulations in GROMACS, a 2 fs time step was used, and the coordinates were recorded every picosecond for further analysis.

### 4. Conclusion

The current study explores a computational approach for finding host-derived miRNAs that could silence the mRNA of the RSV virus. Results predicted that two miRNAs, hsa_miRNA-2278 and hsa_miRNA-6732-3p, established good binding complementarity with the mRNA of RSV. Multiple proteins were identified as active targets for RSV, and the F protein was repeatedly identified as a target in our computational results. Furthermore, miRNA-mRNA docking results showed the binding pattern of hsa_miRNA-2278 and hsa_miRNA-6732-3p to mRNA of F that may have an essential role in the inhibition of the transcriptional activity of the mRNA of F protein. Moreover, the good stability behavior has been depicted in MD simulation. Exploring these small endogenous nucleotides (miRNAs) would be a possible target for better understanding the transcriptional activity of RSV infection.

## Supporting information

**S1 Data. The raw data used for this manuscript has been mentioned in supporting information S1_data.**
(XLSX)

**S1 File. The predicted secondary structure of mRNA has been depicted in S1 File.**
(DOCX)

**S1 Fig. Graphical Abstract.**
(JPG)

## Acknowledgments

AK expresses appreciation for the support received throughout this work. MH also thanks the Steve and Cindy Rasmussen Institute for Genomic Medicine at Nationwide Children's Hospital for providing the opportunity and resources needed to complete this computational research.

## Author contributions

**Conceptualization:** Mubashir Hassan.

**Data curation:** Muhammad Shahzad Iqbal, Zainab Yaseen, Saba Shahzadi.

**Formal analysis:** Zainab Yaseen, Saba Shahzadi.

**Investigation:** Muhammad Shahzad Iqbal.

**Methodology:** Zainab Yaseen.

**Software:** Muhammad Yasir.

**Writing – original draft:** Mubashir Hassan.

**Writing – review & editing:** Wanjoo Chun, Mark E Peeples, Andrzej Kloczkowski.

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
