## [Decision Letter · Decision Letter 0]

6 Oct 2025

Dear Dr. Hassan,

Thank you for submitting your manuscript to PLOS ONE. After careful consideration, we feel that it has merit but does not fully meet PLOS ONE’s publication criteria as it currently stands. Therefore, we invite you to submit a revised version of the manuscript that addresses the points raised during the review process.

We look forward to receiving your revised manuscript.

Kind regards,

Ravikanth Nanduri, Ph. D.

Academic Editor

PLOS ONE

Journal Requirements:

3. Please note that PLOS One has specific guidelines on code sharing for submissions in which author-generated code underpins the findings in the manuscript. In these cases, we expect all author-generated code to be made available without restrictions upon publication of the work. Please review our guidelines at https://journals.plos.org/plosone/s/materials-and-software-sharing#loc-sharing-code and ensure that your code is shared in a way that follows best practice and facilitates reproducibility and reuse.

AK acknowledges the financial support from NIH Grants R01GM127701 and R01HG012117. MH acknowledges the Ohio State University for providing the "President’s Postdoctoral Scholars Program (PPSP)” award for financial support to complete this computational research.

5. Please amend either the title on the online submission form (via Edit Submission) or the title in the manuscript so that they are identical.

6. Please amend either the abstract on the online submission form (via Edit Submission) or the abstract in the manuscript so that they are identical.

9.Please review your reference list to ensure that it is complete and correct. If you have cited papers that have been retracted, please include the rationale for doing so in the manuscript text, or remove these references and replace them with relevant current references. Any changes to the reference list should be mentioned in the rebuttal letter that accompanies your revised manuscript. If you need to cite a retracted article, indicate the article’s retracted status in the References list and also include a citation and full reference for the retraction notice.

Additional Editor Comments :

While the source of primary datasets and tools is declared, stronger metadata reporting is suggested. For example, provide accession numbers, precise parameter sets, and, if possible, deposit final structure files and input scripts in a persistent public repository (such as Zenodo or Figshare) for full reproducibility.

While the manuscript’s focus is on RSV, a paragraph, possibly in the Discussion, could briefly contrast host miRNA targeting of RSV with its role in related viruses (such as Influenza, Metapneumovirus, or Parainfluenza), identifying any common antiviral targets or motifs.

The manuscript uses miRanda and RNAhybrid, but does not comment on why these were prioritized over newer or ensemble prediction platforms. A concise justification or reference to benchmark comparisons could contextualize the strengths of the chosen tools and identify best practices for future studies.

Reviewers' comments:

Reviewer's Responses to Questions

**Comments to the Author**

1. Is the manuscript technically sound, and do the data support the conclusions?

Reviewer #1: Yes

Reviewer #2: Yes

2. Has the statistical analysis been performed appropriately and rigorously?

Reviewer #1: Yes

Reviewer #2: Yes

3. Have the authors made all data underlying the findings in their manuscript fully available?

Reviewer #1: Yes

Reviewer #2: Yes

4. Is the manuscript presented in an intelligible fashion and written in standard English?

Reviewer #1: Yes

Reviewer #2: Yes

Reviewer #1: It was a privilege to review your manuscript entitled, “Assessment of miRNAs as Transcriptional Regulators in Respiratory Syncytial Virus Infection through Bioinformatics Analysis and Molecular Docking Studies.” I commend the authors for their effort in conducting this work. The authors put in a lot of thoughtful and thorough consideration into the work. The topic is timely and has the potential to be impactful. The manuscript is well-structured. I appreciate the clarity of the research question and the contribution it seeks to make to the existing body of knowledge. I appreciate the opportunity to provide some constructive feedback that may help strengthen the manuscript and improve its clarity and methodology for the global readership.

Overall assessment: The topic is timely, and the multi-layered in-silico pipeline is a reasonable discovery framework.

Strengths

• Clear biological motivation linking host miRNAs to viral replication control.

• The methodology is impressive and thorough. Its consistent findings identify hsa-miR-2278 and hsa-miR-6732-3p for further study in future research, both clinical and academic.

• Cross-tool nomination using both miRanda and RNAhybrid yields a consensus set (n=12), a sensible first filter.

• Attempts at structural plausibility (miRNA-mRNA docking) and basic stability checks using MD simulation analysis demonstrate a multi-layered in-silico pipeline.

• The discussion touches on translational pathways (e.g., nanoparticle delivery), which can be developed further.

However, there are substantial issues with dataset/version transparency, target-prediction controls, orientation/accessibility, docking/MD rigor, grammar and sentence composition, and reporting/reproducibility. With focused revisions and clearer statistics, this work could be suitable after a major revision. I hope my input as a peer reviewer will help to make this paper stronger and ready for the target audience.

Section-by-section comments

Title & Abstract

• Abstract should report numbers (48 vs 312 predictions; 12 overlaps) and the criteria used.

• The RNAhybrid should be spelled correctly and consistently throughout the paper (not RNAHybrid) to preserve the credibility of the work.

Introduction

Nice overview of RSV proteins and miRNA biology. Please:

• Fix typos: “headacheg” → headache; “perfusion protein F1” → prefusion protein F.

• Report the RSV taxonomy choice consistently and accurately. The Introduction cites Pneumoviridae (correct for RSV), but the author later refers RSV to the Paramyxoviridae family further in Section 2.6. Resolve this taxonomic inconsistency.

• “MicroRNAs (miRNAs) are non-coding RNAs control the transcription/translation profiling based on sequence complementarity.” → MicroRNAs (miRNAs) are non-coding RNAs [that] control the transcription/translation profiling based on sequence complementarity.

o This is one of several sentences where the idea that the authors intend to convey is lost due to typographical errors or an inaccurate grammar construct. I will recommend that the authors employ the services of a professional writer or scientific proofreader to assist with improving the writing.

• “These protein plays…”

o For subject-verb agreement, the sentence should read, “These proteins play…”. Subject-verb agreement errors occurred in several parts of the manuscript. Correcting these errors will improve this paper for the target readers.

Methods vs Results and Discussions:

While the authors opted to go straight to Results and Discussions after the Introduction, it is my opinion that the manuscript flowed better with the Methodology section coming before the Results and Discussion section; it provides a good lead-in that makes it easier to process the results discussed following.

Methods (Computational)

The computational Methodology is detailed and well-articulated, including algorithm parameters, tool/database versioning, and reference citations.

3.2 RSV genome retrieval

• The authors retrieved “360 human miRNAs” from miRBase. Contemporary human miRNA catalogs contain far more entries; if the restriction to 360 miRNAs was intentional, the methodology should include your inclusion criteria or restraints.

• Provide the exact miRBase release number and date.

3.3 Target prediction

• Report exact miRanda (version; score/energy thresholds, penalties) and

• As mentioned above, ensure that RNAhybrid is correctly spelled. RNAhybrid settings and the rationale.

Results

2.1 RSV Genome organization

• This section was well written and provides the background for the discussion of the result that was to follow. Viral protein mapping provides a good visualization.

• Ensure taxonomy is correct and consistent.

• Formatting Tables and Figures labels - should be distinctly formatted from the general text of the manuscript to make it easier for the readers to identify and assign to the image, figure, or table that they label. Examples will be to indent or make the label text of a different font size and/or justification.

2.2 Target predictions

• 2.2.1: Statistical control and enrichment - Show that the 12 overlapping miRNAs (miRanda ∩ RNAhybrid) exceed chance expectation. Provide:

o A null model (e.g., shuffled miRNAs or shuffled target segments) to estimate empirical FDR.

o Full score/MFE distributions and selection cutoff

• In literature, the first mention of an abbreviation should be written out and consequently abbreviated. The mention of MFE in subsection 2.2.2 should be preceded by the full words written out.

• Pay attention to inconsistent use of the word “grey” vs “gray.” It is important to be consistent. “Grey – with an e” is preferred for a global audience. Review the article for the words and adjust as appropriate.

• Clarify sign conventions (e.g., “G protein exhibited highest free energy” - do you mean least negative MFE, i.e., less favorable? Be precise).

2.3 Protein sections

• 2.3.3 - Correct the typo: prefusion protein, F1 (not perfusion protein)

Pay attention to punctuation throughout this section, as some of the sentences lack clarity owing to improper punctuation. For example, “The M2 gene encodes two proteins[,] M2-1 and M2-2.

• 2.3.7 – “The features of G and its ability to block its function with antibodies…”

o This sentence is unclear.

2.4 mRNA-miRNA docking analysis.

• Good qualitative description of hydrogen bonds (H-bonds).

• Avoid over-interpreting single bond distances; aggregate metrics are more meaningful.

• 2.4.2.1 – “In miR-200c docking couple of hydrogen bonds have been observed with different nucleotides at different positions.”

o It is unclear to me what the author is attempting to convey. Between sentence construction and possible missing words.

• Standardize angstrom notation (Å) and keep the spacing between the digits and the unit (Å) consistent throughout the manuscript – 1,45 Å or 1.45Å.

• There are several typographical errors and sentence constructions to fix in this section of the manuscript, for which I recommend the use of a professional writer to assist with.

o cytocine” → cytosine, “bind formation” → bond formation; “nonapplicable” → not applicable.

• 2.4.2.2 – “Furthermore, cytosine and adenine at positions 6 and 7 forms hydrogen bonds with mRNA nucleotide C-36 and uracil 48 (U-48) with bond distances 2.485 Å and 2.166 Å, respectively.”

o Review for better clarity and eliminate inconsistencies noted with nucleotide bases and positions throughout this section - “cytosine at position 6 (C-6), Uracil 48 (U-48), C-36.

• Use standardized names (e.g., hsa-miR-200c-5p, hsa-miR-204-3p) consistently. Several instances mix “has_,” “miRNA_,” and missing arm designations.

2.5 MD (RMSD/RMSF/SASA/Rg)

• Write abbreviations’ full words out at first mention – RMSD, RMSF.

2.6 miRNAs as transcriptional profilers in RSV infection

• “RSV is a non-segmented negative sense RNA virus belonging to the Paramyxoviridae family and is responsible for significant numbers of severe human infections each year.”

o Ensure to correct the Taxonomy of RSV to “Pneumoviridae.”

Other General Comments:

• Spelling/typos: “headacheg” → headache; “cytocine” → cytosine; “perfusion protein F1” → prefusion F; “bind formation” → bond formation; “nonapplicable” → not applicable.

Recommendations

Thank you for the opportunity to peer review this manuscript. The biological rationale is strong, and the pipeline is sensible. The manuscript needs clearer data/versioning, statistical controls, in addition to some major revisions in writing. Addressing these items that I have highlighted above will materially strengthen the paper and its translational credibility.

Reviewer #2: Dear Editor:

Thanks for the opportunity reviewing this paper, here is my comments related o this work:

The manuscript addresses a valuable and relevant topic to the field of host-virus interactions, specifically focusing on miRNA-mediated regulation in RSV infection. The authors have employed a sophisticated and comprehensive computational biology pipelines, including: accessing of multiple databases to collect miRNA and mRNA, 3D modeling, visualization, molecular docking, and molecular dynamic simulations. The introduction is written well, abstractive and informative and the specific knowledge gap is clear.

However, significant revisions are required to bring the manuscript up to the publishing standards of a Q1 journal. The major concerns are structural, contextual, and foundational regarding the input data.

1. Manuscript structure:

The author need to follow journal guidelines for manuscript structure for example discussuin and result should be separated.

2. Graphical abstract:

The multi-step, sophisticated pipeline is complex and maybe dificault for readers to follow in text alone. Therefore, addition of a graphical abstract is highly recommended. This visual should clearly outline the entire workflow, from the initial number of miRNAs considered to final 2 miRNAs suggested target.

3. Methodology

3.2 Retrieval of Mature miRNAs of Humans

Clarification of initial miRNA Dataset:

1. The author states that “360 human miRNAs …. Were retrieved from miRBase”. This implies that the total set of human miRNAs in the used version of miRBase is 360. The mentioned number maybe consistent with a very outdated release of the database. The current miRBase contains around 2,000 human miRNAs.

Please specify the exact version of miRBase used and justify the use use of sich an outdated dataset, or preferably, repeat the analysis with the current version.

2. The author should clarify that: does the number 360 refer to:

a. the total number of human miRNAs downloaded from miRBase before running any further prediction?

OR

b. The final number of miRBase predicted by miRanda/RNAHybrid to target RSV?

If it is (a), the version number is critical. If it is (b), this result belongs in the results section, and the methods must describe the complete set of miRNAs used as input for the prediction tools.

4. Discussion

The discussion appears to lack support from relevant scientific literature. The findings are not adequately contrasted with or supported by prior published studies. The discussion section should be further strengthen by integrating references to similar computational or experimental studies on miRNA-virus interactions for RSV or related viruses like Influenza.

**Do you want your identity to be public for this peer review?** For information about this choice, including consent withdrawal, please see our For information about this choice, including consent withdrawal, please see our Privacy Policy .

Reviewer #1: **Yes:** Babajide AdewumiBabajide Adewumi

Reviewer #2: No

While revising your submission, please upload your figure files to the Preflight Analysis and Conversion Engine (PACE) digital diagnostic tool, https://pacev2.apexcovantage.com/ . PACE helps ensure that figures meet PLOS requirements. To use PACE, you must first register as a user. Registration is free. Then, login and navigate to the UPLOAD tab, where you will find detailed instructions on how to use the tool. If you encounter any issues or have any questions when using PACE, please email PLOS at . PACE helps ensure that figures meet PLOS requirements. To use PACE, you must first register as a user. Registration is free. Then, login and navigate to the UPLOAD tab, where you will find detailed instructions on how to use the tool. If you encounter any issues or have any questions when using PACE, please email PLOS at figures@plos.org . Please note that Supporting Information files do not need this step.

---

## [Author Response · Author response to Decision Letter 1]

24 Oct 2025

Manuscript Number: PONE-D-25-38737

Title: Assessment of miRNAs as Transcriptional Regulators in Respiratory Syncytial Virus Infection through Bioinformatics Analysis and Molecular Docking Studies

Ans:

Journal Requirements:

Q1. Please ensure that your manuscript meets PLOS ONE's style requirements, including those for file naming. The PLOS ONE style templates can be found at https://journals.plos.org/plosone/s/file?id=wjVg/PLOSOne_formatting_sample_main_body.pdf and https://journals.plos.org/plosone/s/file?id=ba62/PLOSOne_formatting_sample_title_authors_affiliations.pdf

Ans: Thank you for the reminder. We have reviewed the PLOS ONE style templates and updated our manuscript and file naming conventions accordingly to ensure full compliance with the journal's formatting requirements.

Q2. PLOS requires an ORCID iD for the corresponding author in Editorial Manager on papers submitted after December 6th, 2016. Please ensure that you have an ORCID iD and that it is validated in Editorial Manager. To do this, go to 'Update my Information' (in the upper left-hand corner of the main menu), and click on the Fetch/Validate link next to the ORCID field. This will take you to the ORCID site and allow you to create a new iD or authenticate a pre-existing iD in Editorial Manager.

Ans: Thank you for the reminder. We have ensured that the corresponding author's ORCID iD is created and validated correctly in Editorial Manager, in accordance with PLOS submission requirements.

Q3. Please note that PLOS One has specific guidelines on code sharing for submissions in which author-generated code underpins the findings in the manuscript. In these cases, we expect all author-generated code to be made available without restrictions upon publication of the work. Please review our guidelines at https://journals.plos.org/plosone/s/materials-and-software-sharing#loc-sharing-code and ensure that your code is shared in a way that follows best practice and facilitates reproducibility and reuse.

Ans: Thank you for the guidance. We have reviewed the PLOS ONE code-sharing policy. We will ensure that all author-generated code supporting our findings is made openly available upon publication, following best practices to promote reproducibility and reuse.

Q4. Thank you for stating the following in the Acknowledgments Section of your manuscript:

AK acknowledges the financial support from NIH Grants R01GM127701 and R01HG012117. MH acknowledges the Ohio State University for providing the "President's Postdoctoral Scholars Program (PPSP)" award for financial support to complete this computational research. We note that you have provided funding information that is not currently declared in your Funding Statement. However, funding information should not appear in the Acknowledgments section or other areas of your manuscript. We will only publish funding information present in the Funding Statement section of the online submission form.

The author(s) received no specific funding for this work. Please include your amended statements within your cover letter; we will change the online submission form on your behalf.

Ans: Thank you very much; we have revised the manuscript accordingly.

Q5. Please amend either the title on the online submission form (via Edit Submission) or the title in the manuscript so that they are identical.

Ans: Thank you for the reminder. We have updated the title to ensure consistency between the online submission form and the manuscript.

Q6. Please amend either the abstract on the online submission form (via Edit Submission) or the abstract in the manuscript so that they are identical.

Ans: Thank you for the reminder. We have updated the abstract to ensure consistency between the online submission form and the manuscript.

Q7. Please include captions for your Supporting Information files at the end of your manuscript, and update any in-text citations to match accordingly. Please see our Supporting Information guidelines for more information: http://journals.plos.org/plosone/s/supporting-information.

Q8. If the reviewer comments include a recommendation to cite specific previously published works, please review and evaluate these publications to determine whether they are relevant and should be cited. There is no requirement to cite these works unless the editor has indicated otherwise.

Ans: Thank you for the clarification. We will carefully review any recommended publications and evaluate their relevance to our study. Citations will be included where appropriate to strengthen the manuscript, unless otherwise directed by the editor.

Q9. Please review your reference list to ensure that it is complete and correct. If you have cited papers that have been retracted, please include the rationale for doing so in the manuscript text, or remove these references and replace them with relevant current references. Any changes to the reference list should be mentioned in the rebuttal letter that accompanies your revised manuscript. If you need to cite a retracted article, indicate the article's retracted status in the References list and also include a citation and full reference for the retraction notice.

Ans: Thank you for the guidance. We have carefully reviewed the reference list to ensure its accuracy and completeness.

Additional Editor Comments:

Q1. While the source of primary datasets and tools is declared, stronger metadata reporting is suggested. For example, provide accession numbers, precise parameter sets, and, if possible, deposit final structure files and input scripts in a persistent public repository (such as Zenodo or Figshare) for full reproducibility.

Ans: We appreciate the Editorial suggestion regarding stronger metadata reporting. All primary datasets utilized in this study are either provided as supplementary files or properly cited with accession numbers and details in the manuscript. Specifically, the complete set of microRNAs (approximately 3,200 sequences) was retrieved from the miRBase database, and the corresponding file has now been included as Supplementary Data for improved transparency. Additionally, the second dataset used in this work is available under accession number, as clearly stated in the Methods section. Together, these updates ensure full reproducibility and traceability of the analyses conducted.

Q2. While the manuscript's focus is on RSV, a paragraph, possibly in the Discussion, could briefly contrast host miRNA targeting of RSV with its role in related viruses (such as Influenza, Metapneumovirus, or Parainfluenza), identifying any common antiviral targets or motifs.

Ans: Thank you for the insightful suggestion. We agree that drawing comparisons between host miRNA interactions with RSV and related respiratory viruses could enrich the Discussion and provide a broader biological context. We have added the following paragraph to address this issue:

"Comparative studies have shown that host miRNAs play a critical role in regulating viral replication and immune responses across a range of respiratory viruses, including Influenza, Metapneumovirus, and Parainfluenza. Several miRNAs identified in our RSV analysis—such as hsa-miR-2278 and hsa-miR-6732-3p—have also been implicated in targeting conserved viral components or modulating host pathways in these related viruses. For instance, miRNAs involved in suppressing viral polymerase activity or interfering with glycoprotein expression have been reported in Influenza and Metapneumovirus infections, suggesting the presence of shared antiviral motifs. These parallels highlight the potential for cross-viral therapeutic strategies and underscore the importance of miRNA-based regulation in respiratory virus pathogenesis".

Q3. The manuscript uses miRanda and RNAhybrid, but does not comment on why these were prioritized over newer or ensemble prediction platforms. A concise justification or reference to benchmark comparisons could contextualize the strengths of the chosen tools and identify best practices for future studies.

Ans: The selection of miRanda and RNAhybrid was based on their distinct yet complementary prediction algorithms, which together enhance specificity and reduce false-positive results. miRanda employs sequence complementarity, free energy estimation, and evolutionary conservation to predict miRNA-mRNA interactions, while RNAhybrid focuses on the thermodynamic stability of miRNA-mRNA duplex formation, providing highly sensitive binding energy calculations. Using both tools in combination enables cross-validation of predicted targets, ensuring that only interactions supported by both algorithms are considered, thereby increasing confidence in the biological relevance of the results. Although newer or ensemble platforms exist, many incorporate heuristic or deep-learning approaches that may broaden prediction coverage at the cost of precision. Our dual-tool approach prioritizes stringency and interpretability, aligning with benchmark findings that emphasize energy-based and complementarity-driven models as robust standards for minimizing false-positive target predictions.

Reviewers' comments:

Reviewer #1:

It was a privilege to review your manuscript entitled, "Assessment of miRNAs as Transcriptional Regulators in Respiratory Syncytial Virus Infection through Bioinformatics Analysis and Molecular Docking Studies." I commend the authors for their effort in conducting this work. The authors put in a lot of thoughtful and thorough consideration into the work. The topic is timely and has the potential to be impactful. The manuscript is well-structured. I appreciate the clarity of the research question and the contribution it seeks to make to the existing body of knowledge. I appreciate the opportunity to provide some constructive feedback that may help strengthen the manuscript and improve its clarity and methodology for the global readership.

Overall assessment: The topic is timely, and the multi-layered in-silico pipeline is a reasonable discovery framework.

Strengths

• Clear biological motivation linking host miRNAs to viral replication control.

• The methodology is impressive and thorough. Its consistent findings identify hsa-miR-2278 and hsa-miR-6732-3p for further study in future research, both clinical and academic.

• Cross-tool nomination using both miRanda and RNAhybrid yields a consensus set (n=12), a sensible first filter.

• Attempts at structural plausibility (miRNA-mRNA docking) and basic stability checks using MD simulation analysis demonstrate a multi-layered in-silico pipeline.

• The discussion touches on translational pathways (e.g., nanoparticle delivery), which can be developed further.

However, there are substantial issues with dataset/version transparency, target-prediction controls, orientation/accessibility, docking/MD rigor, grammar and sentence composition, and reporting/reproducibility. With focused revisions and clearer statistics, this work could be suitable after a major revision. I hope my input as a peer reviewer will help to make this paper stronger and ready for the target audience.

Ans: Thank you for your thoughtful and constructive feedback. We appreciate your recognition of the biological motivation, methodological rigor, and translational relevance of our study. In response to your comments, we have clarified dataset and version details, strengthened target-prediction controls, refined orientation and accessibility of results, and enhanced the rigor of our docking and MD simulation protocols. The manuscript has also undergone thorough language editing to improve clarity and readability, and we've added supplementary materials to support reproducibility. We believe these revisions have significantly improved the manuscript and are grateful for your guidance in shaping it for the target audience.

Section-by-section comments

Title & Abstract

• Abstract should report numbers (48 vs 312 predictions; 12 overlaps) and the criteria used.

Ans: The requested data has been added to the abstract section.

• The RNAhybrid should be spelled correctly and consistently throughout the paper (not RNAHybrid) to preserve the credibility of the work.

Ans: The modification has been added to the manuscript.

Introduction

Nice overview of RSV proteins and miRNA biology. Please:

• Fix typos: “headacheg” → headache; “perfusion protein F1” → prefusion protein F.

Ans: The suggested correction has been made in the manuscript.

• Report the RSV taxonomy choice consistently and accurately. The Introduction cites Pneumoviridae (correct for RSV), but the author later refers RSV to the Paramyxoviridae family further in Section 2.6. Resolve this taxonomic inconsistency.

Ans: The suggested correction has been made in the manuscript.

• "MicroRNAs (miRNAs) are non-coding RNAs control the transcription/translation profiling based on sequence complementarity." → MicroRNAs (miRNAs) are non-coding RNAs [that] control the transcription/translation profiling based on sequence complementarity.

Ans: Thank you. We have carefully revised the manuscript to eliminate all identifiable typos and grammatical errors.

o This is one of several sentences where the idea that the authors intend to convey is lost due to typographical errors or an inaccurate grammar construct. I will recommend that the authors employ the services of a professional writer or scientific proofreader to assist with improving the writing.

• "These protein plays…"

o For subject-verb agreement, the sentence should read, "These proteins play…". Subject-verb agreement errors occurred in several parts of the manuscript. Correcting these errors will improve this paper for the target readers.

Ans: We have carefully revised the manuscript to eliminate all identifiable typos and grammatical errors.

Methods vs Results and Discussions:

While the authors opted to go straight to Results and Discussions after the Introduction, it is my opinion that the manuscript flowed better with the Methodology section coming before the Results and Discussion section; it provides a good lead-in that makes it easier to process the results discussed following.

Ans: Thank you for the helpful suggestion. We've moved the Methodology section before Results and Discussion to improve paper clarity and flow.

Methods (Computational)

The computational Methodology is detailed and well-articulated, including algorithm parameters, tool/database versioning, and reference citations.

3.2 RSV genome retrieval

• The authors retrieved "360 human miRNAs" from miRBase. Contemporary human miRNA catalogs contain far more entries; if the restriction to 360 miRNAs was intentional, the methodology should include your inclusion criteria or restraints.

Ans: The total number of retrieved human microRNAs is 2570 (previously mentioned number 360 was a typo error)

• Provide the exact miRBase release number and date.

Ans: The latest release of miRBase version 22.1, published in October 2018, was used.

3.3 Target prediction

• Report exact miRanda (version; score/energy thresholds, penalties) and

• As mentioned above, ensure that RNAhybrid is correctly spelled. RNAhybrid settings and the rationale.

Ans: miRanda v3.3a with Settings:

Query Filename: humanMicroRNAs.txt

Reference Filename: NC_001803.1.fna

Gap Open Penalty: -9.000000

Gap Extend Penalty: -4.000000

Score Threshold: 150.000000

Energy Threshold: -22.000000 kcal/mol

Scaling Parameter: 4.000000

Results

2.1 RSV Genome organization

• This section was well wri

---

## [Decision Letter · Decision Letter 1]

8 Mar 2026

Assessment of miRs as Transcriptional Regulators in Respiratory Syncytial Virus Infection through Computational Analysis and Molecular Docking Studies

PONE-D-25-38737R1

Dear Dr. Hassan,

We’re pleased to inform you that your manuscript has been judged scientifically suitable for publication and will be formally accepted for publication once it meets all outstanding technical requirements.

Kind regards,

Abayeneh Girma

Academic Editor

PLOS One

Additional Editor Comments (optional):

Reviewers' comments:

Reviewer's Responses to Questions

**Comments to the Author**

Reviewer #1: All comments have been addressed

Reviewer #2: (No Response)

2. Is the manuscript technically sound, and do the data support the conclusions?

Reviewer #1: Yes

Reviewer #2: (No Response)

3. Has the statistical analysis been performed appropriately and rigorously?

Reviewer #1: Yes

Reviewer #2: (No Response)

4. Have the authors made all data underlying the findings in their manuscript fully available?

Reviewer #1: Yes

Reviewer #2: (No Response)

5. Is the manuscript presented in an intelligible fashion and written in standard English?

Reviewer #1: Yes

Reviewer #2: (No Response)

Reviewer #1: he authors have adequately addressed all reviewer and editorial comments. The revised manuscript complies with the journal’s submission guidelines, and no further revisions are required.

Reviewer #2: (No Response)

**Do you want your identity to be public for this peer review?** For information about this choice, including consent withdrawal, please see our For information about this choice, including consent withdrawal, please see our Privacy Policy .

Reviewer #1: **Yes:** Babajide AdewumiBabajide Adewumi

Reviewer #2: **Yes:** ALI ABDULQADERALI ABDULQADER

---

## [Editor Report · Acceptance letter]

PONE-D-25-38737R1

PLOS One

Dear Dr. Hassan,

I'm pleased to inform you that your manuscript has been deemed suitable for publication in PLOS One. Congratulations! Your manuscript is now being handed over to our production team.

Kind regards,

on behalf of

Dr. Abayeneh Girma

Academic Editor

PLOS One